# DISTRIBUTED BANDIT LEARNING: NEAR-OPTIMAL REGRET WITH EFFICIENT COMMUNICATION

**Yuanhao Wang**[*]
Institute for Interdisciplinary Information Sciences,
Tsinghua University
yuanhao-16@mails.tsinghua.edu.cn

**Jiachen Hu**[*]
School of Electronics Engineering and Computer Science,
Peking University
NickH@pku.edu.cn

**Xiaoyu Chen**
Key Laboratory of Machine Perception, MOE, School of EECS,
Peking University
cxy30@pku.edu.cn

**Liwei Wang**
Key Laboratory of Machine Perception, MOE, School of EECS
Center for Data Science, Peking University
wanglw@cis.pku.edu.cn

## ABSTRACT

We study the problem of regret minimization for distributed bandits learning, in which $M$ agents work collaboratively to minimize their total regret under the coordination of a central server. Our goal is to design communication protocols with near-optimal regret and little communication cost, which is measured by the total amount of transmitted data. For distributed multi-armed bandits, we propose a protocol with near-optimal regret and only $O(M \log(MK))$ communication cost, where $K$ is the number of arms. The communication cost is independent of the time horizon $T$, has only logarithmic dependence on the number of arms, and matches the lower bound except for a logarithmic factor. For distributed $d$-dimensional linear bandits, we propose a protocol that achieves near-optimal regret and has communication cost of order $O\left((Md + d \log \log d) \log T\right)$, which has only logarithmic dependence on $T$.

## 1 INTRODUCTION

Bandit learning is a central topic in online learning, and has various real-world applications, including clinical trials (Wang, 1991), model selection (Maron & Moore, 1994) and recommendation systems (Agarwal et al., 2009; Li et al., 2010; Abe et al., 2003). In many tasks using bandit algorithms, it is appealing to employ more agents to learn collaboratively and concurrently in order to speed up the learning process. In many other tasks, the sequential decision making is distributed by nature. For instance, multiple spatially separated labs may be working on the same clinical trial. In such distributed applications, communication between agents is critical, but may also be expensive or time-consuming. Another example is a recommendation system deployed on multiple servers to handle high demand. Since the communication between servers may cause service latency, it would be desirable to design communication strategies without communicating too much. This motivates us to consider efficient protocols for distributed learning in bandit problems.

A straightforward communication protocol for bandit learning is *immediate sharing*: each agent shares every new sample immediately with others. Under this scheme, agents can have good collaborative behaviors close to that in a centralized setting. However, the amount of communicated data is directly proportional to the total size of collected samples. When the bandit is played for a long timescale, the cost of communication would render this scheme impractical. A natural question to ask is: How

---

[*]These two authors contributed equally.

much communication is actually needed for near-optimal performance? In this work, we show that the answer is somewhat surprising: The required communication cost has almost *no* dependence on the time horizon.

In this paper, we consider the distributed learning of stochastic multi-armed bandits (MAB) and stochastic linear bandits. There are $M$ agents interacting with the same bandit instance in a synchronized fashion. In time steps $t = 1, \cdots, T$, each agent pulls an arm and observes the associated reward. Between time steps, agents can communicate via a server-agent network. Following the typical formulation of single-agent bandit learning, we consider the task of regret minimization (Lai et al., 1987; Dani et al., 2008; Bubeck et al., 2012). The total regret of all agents is used as the performance criterion of a communication protocol. The communication cost is measured by the total amount of data communicated in the network. Our goal is to minimize communication cost while maintaining near-optimal performance, that is, regret comparable to the optimal regret of a single agent in $MT$ interactions with the bandit instance.

For multi-armed bandits, we propose the DEMAB protocol, which achieves near-optimal regret. The amount of transmitted data per agent in DEMAB is independent of $T$, and is logarithmic with respect to other parameters. For linear bandits, we propose the DELB protocol, which achieves near-optimal regret, and has communication cost with at most logarithmic dependence on $T$.

## 1.1 Problem Setting

**Communication Model**    The communication model we consider consists of a server and several agents. Agents can communicate with the server by sending or receiving packets. Each data packet contains integers or real numbers. We define the communication cost of a protocol as the number of integers or real numbers communicated between server and agents[1]. We assume that communication between server and agents has zero latency. Note that protocols in our model can be easily adapted to a network without a server, by designating an agent as the server.

**Distributed Multi-armed Bandits**    In distributed multi-armed bandits, there are $M$ agents, labeled 1,...,$M$. Each agent is given access to the same stochastic $K$-armed bandit instance. Each arm $k$ in the instance is associated with a reward distribution $\mathcal{P}_k$. $\mathcal{P}_k$ is supported on $[0, 1]$ with mean $\mu(k)$. Without loss of generality, we assume that arm 1 is the best arm (i.e. $\mu(1) \geq \mu(k)$, $\forall k \in [K]$). At each time step $t = 1, 2, ..., T$, each agent $i$ chooses an arm $a_{t,i}$, and receives reward $r_{t,i}$ independently sampled from $\mathcal{P}_{a_{t,i}}$. The goal of the agents is to minimize their total regret, which is defined as

$$REG(T) = \sum_{t=1}^{T} \sum_{i=1}^{M} \left( \mu(1) - \mu(a_{t,i}) \right).$$

**Distributed Linear Bandits**    In distributed linear bandits, the agents are given access to the same $d$-dimensional stochastic linear bandits instance. In particular, we assume that at time step $t$, agents are given an action set $\mathcal{D} \subseteq \left\{ x \in \mathbb{R}^d : \|x\|_2 \leq 1 \right\}$. Agent $i$ chooses action $x_{t,i} \in \mathcal{D}$ and observes reward $y_{t,i}$. We assume that the mean of the reward is decided by an unknown parameter $\theta^* \in \mathbb{R}^d$: $y_{t,i} = x_{t,i}^T \theta^* + \eta_{t,i}$, where $\eta_{t,i} \in [-1, 1]$ are independent and have zero mean. For simplicity, we assume $\|\theta^*\|_2 \leq 1$. For distributed linear bandits, the cumulative regret is defined as the sum of individual agent's regrets:

$$REG(T) = \sum_{t=1}^{T} \sum_{i=1}^{M} \left( \max_{x \in \mathcal{D}} x^T \theta^* - x_{t,i}^T \theta^* \right).$$

Here, we assume that the action set is fixed. A more general setting considers a time-varying action set $\mathcal{D}_t$. In both cases, algorithms with $O(d\sqrt{T} \log T)$ regret have been proposed (Abbasi-Yadkori et al., 2011), while a regret lower bound of $\Omega\left(d\sqrt{T}\right)$ is shown in Dani et al. (2008).

For both distributed multi-armed bandits and distributed linear bandits, our goal is to use as little communication as possible to achieve near-optimal regret. Since any $M$-agent protocol running for

---

[1]In our protocols, the number of bits each integer or real number uses is only logarithmic w.r.t. instance scale. Using the number of bits as the definition of communication complexity instead will only result in an additional logarithmic factor. The number of communicated bits is analyzed at the end of Appendix C G and H

| Setting | Algorithm | Regret | Communication |
|---------|-----------|--------|---------------|
| Multi-armed bandits | Immediate Sharing | $O\left(\sqrt{MKT}\right)$ | $O(M^2T)$ |
| | DEMAB (Sec. 3.2) | $O\left(\sqrt{MKT\log(MK)}\right)$ | $O(M\log(MK))$ |
| | Lower bound (Sec. 3.4) | $o\left(M\sqrt{KT}\right)$ | $\Omega(M)$ |
| Linear bandits | Immediate Sharing | $O\left(d\sqrt{MT\log T}\right)$ | $O(M^2dT)$ |
| | DCB (Korda et al., 2016) | $O\left(d\sqrt{MT}\log T\right)$ | $O\left(Md^2T\right)$ |
| | DELB (Sec. 4.2) | $O\left(d\sqrt{MT\log T}\right)$ | $O\left((Md + d\log\log d)\log T\right)$ |
| | DisLinUCB (Sec. 4.4) | $O\left(d\sqrt{MT}\log^2 T\right)$ | $O\left(M^{1.5}d^3\right)$ |

Table 1: Summary of baseline approaches and our results

$T$ steps can be simulated by a single-agent bandit algorithm running for $MT$ time steps, the regret of any protocol is lower bounded by the optimal regret of a single-agent algorithm running for $MT$ time steps. Therefore, we consider $\tilde{O}(\sqrt{MKT})$ regret for multi-armed bandits and $\tilde{O}(d\sqrt{MT})$ regret for linear bandits to be near-optimal.

We are mainly interested in the case where the time horizon $T$ is the dominant factor (compared to $M$ or $K$). Unless otherwise stated, we assume that $T > \max\{\frac{M\log^2 M}{K}, M, K\}$ in the multi-armed bandits case and $T > M$ in the linear bandits case.

## 1.2 Our Contribution

Now we give an overview of our results. In both settings, we present communication-efficient protocols that achieve near-optimal regret. Our results are summarized in Table 1.

Our results are compared with a naive baseline solution called *immediate sharing* in Table 1: each agent sends the index of the arm he pulled and the corresponding reward he received to every other agent via the server immediately. This protocol can achieve near-optimal regret for both MAB and linear bandits ($O(\sqrt{MKT})$ and $\tilde{O}(d\sqrt{MT})$), but comes with high communication cost ($O(M^2T)$ and $O(M^2dT)$).

**Distributed MAB**  For distributed multi-armed bandits, we propose DEMAB (Distributed Elimination for MAB) protocol, which achieves near-optimal regret ($O\left(\sqrt{MKT\log(MK)}\right)$) with $O(M\log(MK))$ communication cost. The communication cost is independent of the number of time steps $T$ and grows only logarithmically w.r.t. the number of arms. We also prove the following lower bound: For any protocol with regret bound $o(M\sqrt{KT})$, the expected communication cost is at least $\Omega(M)$. That is, in order to achieve near-optimal regret, the communication cost of DEMAB matches our lower bound except for logarithmic factors.

**Distributed Linear Bandits**  We propose DELB (Distributed Elimination for Linear Bandits), an elimination based protocol for distributed linear bandits which achieves near-optimal regret bound ($O\left(d\sqrt{MT\log T}\right)$) with communication cost $O\left((Md + d\log\log d)\log T\right)$. The communication cost of DELB enjoys nearly linear dependence on both $M$ and $d$, and has at most logarithmic dependence on $T$. For the more general case where the action set is time-varying, we propose DisLinUCB (Distributed LinUCB) protocol, which achieves near-optimal regret with $O\left(M^{1.5}d^3\right)$ communication cost.

## 2 Related Work

There has been growing interest in bandits problems with multiple players. One line of research considers the challenging problem of multi-armed bandits with collisions (Rosenski et al., 2016;

Bistritz & Leshem, 2018; Kalathil et al., 2014), in which the reward for an arm is 0 if it is chosen by more than one player. The task is to minimize regret without communication. Their setting is motivated by problems in cognitive radio networks, and is fairly different from ours.

Szörényi et al. (2013) and Korda et al. (2016) consider the distributed learning of MAB and linear bandits with restriction on the communication network. Motivated by fully decentralized applications, Szörényi et al. (2013) consider P2P communication networks, where an agent can communicate with only two other agents at each time step. A gossip-based $\epsilon$-greedy algorithm is proposed for distributed MAB. Their algorithm achieves a speedup linear in $M$ in terms of error rate, but the communication cost is linear in $T$. The work of Korda et al. (2016) uses a gossip protocol for regret minimization in distributed linear bandits. The main difference between their setting and ours is that each agent is only allowed to communicate with one agent at each time step in Korda et al. (2016) [2]. Their algorithm achieves near-optimal $\left( O\left( d\sqrt{MT}\log T \right) \right)$ total regret using $O(Md^2T)$ communication cost.

Another setting in literature concerns about distributed pure exploration in multi-armed bandits (Hillel et al., 2013; Tao et al., 2019), where the communication model is the most similar one to ours. These works use elimination based protocols for collaborative exploration, and establish tight bounds for communication-speedup tradeoff. However, their task (speedup in pure exploration) is not directly comparable to ours (i.e. are not reducible to each other). Moreover, in Hillel et al. (2013); Tao et al. (2019), the number of communication rounds is used as the measure of communication, while we use the amount of transmitted data.

## 3 MAIN RESULTS FOR MULTI-ARMED BANDITS

In this section, we first summarize the single-agent elimination algorithm (Auer & Ortner, 2010), and then present our Distributed Elimination for MAB (DEMAB) protocol. The regret and communication efficiency of the protocol is then analyzed in Sec. 3.3. A communication lower bound is presented in Sec. 3.4.

### 3.1 ELIMINATION ALGORITHM FOR SINGLE-AGENT MAB

The elimination algorithm (Auer & Ortner, 2010) is a near-optimal algorithm for single-agent MAB. The agent acts in phases $l = 1, 2, \cdots$, and maintains a set of active arms $\mathcal{A}_l$. Initially, $\mathcal{A}_1 = [K]$ (all arms). In phase $l$, each arm in $\mathcal{A}_l$ is pulled for $\Theta(4^l \log T)$ times; arms with average reward $2^{-l}$ lower than the maximum are then eliminated from $\mathcal{A}_l$.

For each arm $k \in [K]$, define its suboptimality gap to be $\Delta_k := \mu(1) - \mu(k)$. In the elimination algorithm, with high probability arm $k$ will be eliminated after approximately $l_k = \log_2 \Delta_k^{-1}$ phases, in which it is pulled for at most $O\left( \Delta_k^{-2} \log T \right)$ times. It follows that regret is $O\left( \sum_{k:\Delta_k>0} \Delta_k^{-1} \log T \right)$, which is almost instance-optimal. The regret is $O\left( \sqrt{KT \log T} \right)$ in the worst case.

In order to apply elimination algorithm to our setting and remove the $\log T$ dependence in the regret bound, we slightly modify the elimination algorithm: In phase $l$, each arm in $\mathcal{A}_l$ is pulled for $\Theta(4^l \log(4^{-l}T))$ instead of $\Theta(4^l \log(T))$ times. By doing this, the regret bound can be improved from $\sqrt{KT \log T}$ to $\sqrt{KT \log K}$. Our DEMAB protocol is based on this modified elimination algorithm. See Section 3.2 and Appendix C for detailed explanation.

### 3.2 THE DEMAB PROTOCOL

The DEMAB protocol executes in two stages. In the first stage, each agent directly runs the single-agent modified elimination algorithm for $D = \lceil T/MK \rceil$ time steps. The remaining arms of agent $i$ are denoted as $\mathcal{A}^{(i)}$. In $D$ time steps, an agent completes at least $l_0 = \lfloor \log_4 \frac{D}{1000K \log(MK)} \rfloor$ phases. The purpose of this separate burn-in period is to eliminate the worst arms quickly without communication, so that in the second stage, elimination can begin with a small threshold of $O(2^{-l_0})$. $D$ and $l_0$ is chosen so that the total regret within the first stage is $\tilde{O}\left( \sqrt{MT} \right)$.

---

[2]Our algorithms can be modified to meet this restriction with almost no change in performance.

---

**Protocol 1:** Distributed Elimination for Multi-Armed Bandits (DEMAB)

---

1   $D = \lceil T/MK \rceil$, $l_0 = \lfloor \log_4 \left( \frac{D}{1000K \log(MK)} \right) \rfloor$, $m_l = \lfloor 4^{l+3} \log(MKT \cdot 4^{-l}) \rfloor$

    /* Stage 1:  Separate Burn-in                                      */

2   **for** *Agent* $i = 1, \cdots, M$ **do**

3       Agent $i$ runs single-agent elimination for $D$ time steps, denote remaining arms as $\mathcal{A}^{(i)}$

    /* Switching:  Random Allocation                              */

4   Generate public random numbers $r_1,...,r_K$ uniformly distributed in $[M]$

5   $\mathcal{B}_{l_0+1}^{(i)} = \left\{ a \in \mathcal{A}^{(i)} | r_a = i \right\}$, $\mathcal{B}_{l_0+1} = \bigcup_{i \in [M]} \mathcal{B}_{l_0+1}^{(i)}$

    /* Stage 2:  Distributed Elimination                         */

6   **for** $l = l_0 + 1, \cdots$ **do**

7      **if** $N_l = |\mathcal{B}_l| > M$ **then**

8          Agent $i$ sends $n_l^{(i)} = \left| \mathcal{B}_l^{(i)} \right|$ to server; server broadcasts $n_{max} = \max_i n_l^{(i)}$

9          **if** $(n_l^{(1)}, ..., n_l^{(M)})$ *is not balanced* **then**

10              Reallocate   // Adjust $\mathcal{B}_l^{(i)}$ so that their sizes are balanced

11          Agent $i$ pulls each $a \in \mathcal{B}_l^{(i)}$ for $m_l$ times, denotes average reward as $\hat{u}_l(a)$, and then pulls arms in round-robin for $(n_{max} - n_l^{(i)})m_l$ times before the next communication round

12          Communication round: Agent $i$ sends $\max_{a \in \mathcal{B}_l^{(i)}} \hat{u}_l$ to server and waits to receive $u_l^* = \max_{a \in \mathcal{B}_l} \hat{u}_l$ from server

13          Agent $i$ eliminates bad arms: $\mathcal{B}_{l+1}^{(i)} = \left\{ a \in \mathcal{B}_l^{(i)} : \hat{u}_l(a) + 2^{-l} \geq u_l^* \right\}$

14      **else**

15          For each arm in $\mathcal{B}_l$, the server asks $M/N_l$ agents to pull it $m_l N_l/M$ times

16          Server computes $\hat{u}_l(a)$, the average reward for $m_l$ pulls of arm $a$

17          Server eliminates bad arms: $\mathcal{B}_{l+1} = \left\{ a \in \mathcal{B}_l : \hat{u}_l(a) + 2^{-l} \geq \max_{b \in \mathcal{B}_l} \hat{u}_l(b) \right\}$

---

Between the two stages, the remaining arms are randomly allocated to agents. Public randomness is used to allocate the remaining arms to save communication. Agents first generate $(r_1, \cdots, r_K)$, $K$ uniformly random numbers in $[M]$, from a public random number generator. Agent $i$ then computes $\mathcal{B}^{(i)} = \left\{ a \in \mathcal{A}^{(i)} | r_a = i \right\}$. By doing so, agent $i$ keeps each arm in $\mathcal{A}^{(i)}$ with probability $1/M$, and the resulting sets $\mathcal{B}^{(1)}, \cdots, \mathcal{B}^{(M)}$ are disjoint. Meanwhile, every arm in $\bigcap_{i \in [M]} \mathcal{A}^{(i)}$ is kept in $\mathcal{B}_{l_0+1} = \bigcup_{i \in [M]} \mathcal{B}^{(i)}$, so that the best arm remains in $\mathcal{B}_{l_0+1}$ with high probability[3].

In the second stage, agents start to simulate a single-agent elimination algorithm starting from phase $l_0 + 1$. Initially, the arm set is $\mathcal{B}_{l_0+1}$. In phase $l$, each arm in $\mathcal{B}_l$ will be pulled for at least $m_l = \lceil 4^{l+3} \log(MKT \cdot 4^l) \rceil$ times. Denote the average reward of arm $a$ in phase $l$ by $\hat{u}_l(a)$. If $\hat{u}_l(a) < \max_{a' \in \mathcal{B}_l} \hat{u}_l(a') - 2^{-l}$, it will be eliminated; the arm set after the elimination is $\mathcal{B}_{l+1}$.

This elimination in the second stage is performed over $M$ agents in two ways: In *distributed mode* or in *centralized mode*. Let $N_l = |\mathcal{B}_l|$ be the number of remaining arms at the start of phase $l$. If $N_l$ is larger than $M$, the elimination is performed in *distributed mode*. That is, agent $i$ keeps a set of arms $\mathcal{B}_l^{(i)}$, and pulls each arm in $\mathcal{B}_l^{(i)}$ for $m_l$ times in phase $l$. Each agent only needs to send the highest average reward to the server, who then computes and broadcasts $\max_{a \in \mathcal{B}_l} \hat{u}_l(a)$. Agent $i$ then eliminates low-rewarding arms from $\mathcal{B}_l^{(i)}$ on its local copy.

When $N_l \leq M$, the elimination is performed in *centralized mode*. That is, $\mathcal{B}_l$ will be kept and updated by the server[4]. In phase $l$, the server assigns an arm in $\mathcal{B}_l$ to $M/N_l$ agents, and asks each of

---

[3] $\bigcup_{i \in [M]} \mathcal{A}^{(i)}$ may not be a subset of $\mathcal{B}_{l_0+1}$, which is not a problem in the regret analysis.

[4] In the conversion from distributed mode to centralized mode, agents send their local copy to the server. Since the remaining action sets of agents are disjoint, the communication cost is $O(M)$.

them to pull it $m_l N_l / M$ times[5]. The server waits for the average rewards to be reported, and then performs elimination on $\mathcal{B}_l$.

One critical issue here is *load balancing*, especially in distributed mode. Suppose that $n_l^{(i)} = |\mathcal{B}_l^{(i)}|$, $n_{max} = \max_{i \in [M]} n_l^{(i)}$. Then the length of phase $l$ is determined by $n_{max} m_l$. Agent $i$ would need to keep pulling arms for $(n_{max} - n_l^{(i)}) m_l$ times until the start of the next communication round. This will cause an arm to be pulled for much more than $m_l$ times in phase $l$, and can hurt the performance. Therefore, it is vital that at the start of phase $l$, $\vec{n}_l := \left( n_l^{(1)}, ..., n_l^{(M)} \right)$ is balanced[6].

The subroutine `Reallocate` is designed to ensure this by reallocating arms when $\vec{n}_l$ is not balanced. First, the server announces the total number of arms; then, agents with more-than-average number of arms donate surplus arms to the server; the server then distributes the donated arms to the other agents, so that every agent has the same number of arms. However, calling `Reallocate` is communication-expensive: it takes $O\left( \min\{N_l, N_{l'} - N_l\} \right)$ communication cost, where $l$ is the current phase and $l'$ is the last phase where `Reallocate` is called. Fortunately, since $\left\{ \mathcal{B}_{l_0+1}^{(i)} \right\}_{i \in [M]}$ are generated randomly, it is unlikely that one of them contain too many good arms or too many bad arms. By exploiting shared randomness, we greatly reduce the expected communication cost needed for load balancing.

Detailed descriptions of the single-agent elimination algorithm, the `Reallocate` subroutine, and the DEMAB protocol are provided in Appendix B.

Access to a public random number generator, which is capable of generating and sharing random numbers with all agents with negligible communication cost, is assumed in DEMAB. This is not a strong assumption, since it is well known that a public random number generator can be replaced by private random numbers with a little additional communication (Newman, 1991). In our case, only $O\left( M \log T \right)$ additional bits of communication, or $O(M)$ additional communication cost, are required for all of our theoretical guarantees to hold. See Appendix C for detailed discussion.

### 3.3 REGRET AND COMMUNICATION EFFICIENCY OF DEMAB

In this subsection, we show that the DEMAB protocol achieves near-optimal regret with efficient communication, as captured by the following theorem.

**Theorem 1.** *The DEMAB protocol incurs $O\left( \sqrt{MTK \log(MK)} \right)$ regret, $O\left( M \log \frac{MK}{\delta} \right)$ communication cost with probability $1 - \delta$, and $O\left( M \log(MK) \right)$ communication cost in expectation.*

The worst-case regret bound above can be improved to an instance-dependent near-optimal regret bound by changing the choice of $D$ and $l_0$ to 0. In that case the communication cost is $O(M \log T)$, which is a small increase. See Theorem 5 in appendix for detailed discussion.

We now give a sketch of the proof of Theorem 1.

**Regret** In the first stage, each agent runs a separate elimination algorithm for $D$ timesteps, which has regret $O(\sqrt{KD \log(MK)})$. Total regret for all agents in this stage is $O(M\sqrt{KD \log D}) = O(\sqrt{MT \log(MK)})$. After the first stage, each agent must have completed at least $l_0$ phases. Hence, with high probability, before the second stage, $\mathcal{B}_{l_0+1} = \bigcup_{i \in [M]} \mathcal{B}_{l_0+1}^{(i)}$ contains the optimal arm and only arms with suboptimality gap less than $2^{-l_0+1}$.

In the second stage, if $a \in \mathcal{B}_l$, it will be pulled at most $2m_l$ times in phase $l$ because of our load balancing effort. Therefore, if arm $k$ has suboptimality gap $0 < \Delta_k < 2^{-l_0+1}$, it will be pulled for $\tilde{O}\left( \Delta_k^{-2} \right)$ times. It follows that regret in the second stage is $O\left( \sqrt{MKT \log(MK)} \right)$, and that total regret is $O\left( \sqrt{MKT \log (MK)} \right)$.

---

[5]The indivisible case is handled in Appendix B.

[6]By saying a vector of numbers is balanced, we mean the maximum is at most twice the minimum.

**Communication**   In the first stage and the random allocation of arms, no communication is needed. The focus is therefore on the second stage.

During a phase, apart from the potential cost of calling `Reallocate`, communication cost is $O(M)$. The communication cost of calling `Reallocate` in phase $l$ is at most $O(\min\{N_l, N_{l'} - N_l\})$, where $l'$ is the last phase where `Reallocate` is called. Therefore, total cost for calling `Reallocate` in one execution is at most $O(N_{l_1})$, where $l_1$ is the first phase in which `Reallocate` is called. From the definition of $m_l$ and $l_0$, we can see that there are at most $L = O(\log(MK))$ phases in the second stage. Therefore in the worst case, communication cost is $O(ML + K)$ since $N_{l_1} \le K$.

However, in expectation, $N_{l_1}$ is much smaller than $K$. That is to say, `Reallocate` is called for the first time when $N_l$ is much smaller that $K$ with high probability. Since arms are randomly allocated to agents during switching step, when the total number of remaining arms is large enough, $\vec{n}_l$ would be balanced with high probability. In fact, with probability $1 - \delta$, $N_{l_1} = O\left(M \log \frac{MK}{\delta}\right)$. Setting $\delta = 1/K$, we can show that the expected communication complexity is $O(M \log MK)$.

### 3.4   Lower Bound

Intuitively, in order to avoid a $\Omega\left(M\sqrt{KT}\right)$ scaling of regret, $\Theta(M)$ amount of communication cost is necessary; otherwise, most of the agents can hardly do better than a single-agent algorithm. We prove this intuition in the following theorem.

**Theorem 2.** *For any protocol with regret bound $o(M\sqrt{KT})$, the expected communication cost is at least $\Omega(M)$.*

The theorem is proved using a reduction from single-agent bandits to multi-agent bandits, i.e. a mapping from protocols to single-agent algorithms. This theorem shows that, in order to achieve near-optimal regret $\left(\tilde{O}\left(\sqrt{MKT}\right)\right)$, the communication cost is at least $\Omega(M)$. That is to say, the communication cost of DEMAB protocol matches the lower bound except for a logarithmic factor.

## 4   Main Results for Linear Bandits

In this section, we first summarize the single-agent elimination algorithm for linear bandits (Lattimore & Szepesvári, 2019, chap. 22), and then present the Distributed Elimination for Linear Bandits (DELB) protocol. DELB is designed for the case where the action set $\mathcal{D}$ is fixed. Our results for linear bandits with time-varying action set is presented in Sec. 4.4. For convenience, we assume $\mathcal{D}$ is a finite set, which is without loss of generality from a regret point of view[7].

### 4.1   Elimination Algorithm for Single-agent Linear Bandit

The elimination algorithm for linear bandits (Lattimore & Szepesvári, 2019) also iteratively eliminates arms from the initial action set. In phase $l$, the algorithm maintains an active action set $\mathcal{A}_l$. It computes a distribution $\pi_l(\cdot)$ over $\mathcal{A}_l$ and pulls arms according to $\pi_l(\cdot)$. A total of $n_l$ pulls is made in this phase according to $\pi_l(\cdot)$. Then, linear regression is used to estimate the expected reward of each arm based on these pulls. Arms with estimated rewards $\Theta(2^{-l})$ lower than the maximum are eliminated at the end of the phase.

To ensure that arms with suboptimality gap $\Theta(2^{-l})$ are eliminated in phase $l$ with high probability, the estimation error in phase $l$ needs to be smaller than $2^{-l}$. On the other hand, to minimize regret, the number of pulls make in phase $l$ needs to be as small as possible, especially when $l$ is small. Suppose that $V_l(\pi) = \sum_{x \in \mathcal{A}_l} \pi(x)xx^\top$ and $g_l(\pi) = \max_{x \in \mathcal{A}_l} x^\top V_l(\pi)^{-1}x$. By analyzing linear regression (see, e.g., Lattimore & Szepesvári (2019, chap. 21)), one can show that if each arm $x \in \text{Supp}(\pi_l)$ is pulled $\left\lceil \pi(x)g_l(\pi)4^l \log\left(\frac{1}{\delta}\right)\right\rceil$ times, the estimation error for any arm $x \in \mathcal{A}_l$ is at most $2^{-l}$ with high probability. Thus, to minimize $m_l$, one should find a distribution $\pi(\cdot)$ that minimizes $g_l(\pi)$, a task known as $G$-optimal design (Pukelsheim, 2006). It is known that the optimal value for $g_l(\pi)$ is $d$, and

---

[7]When $\mathcal{D}$ is infinite, we can replace $\mathcal{D}$ with an $\epsilon$-net of $\mathcal{D}$, and only take actions in the $\epsilon$-net. If $\epsilon < 1/T$, this will not affect the scaling of the regret. This is a feasible approach, but computationally inefficient.

that there exists a minimizer $\pi^*$ such that the support set of $\pi^*$, also called the core set, has cardinality at most $d(d+1)/2$. Consequently, only $\sum_{x \in \text{Supp}(\pi_l^*)} \lceil \pi^*(x) g_l(\pi^*) 4^l \log(\frac{1}{\delta}) \rceil \leq O(4^l d \log(\frac{1}{\delta}) + d^2)$ pulls are needed in phase $l$.

## 4.2 THE DELB PROTOCOL

In this protocol, we parallelize the data collection part of each phase by sending instructions to agents in a communication-efficient way. In phase $l$, the server and all agents first locally solve the same $G$-optimal design problem on $\mathcal{A}_l$, the remaining set of actions. In our case, only a 2-approximation to the optimal design is needed. That is, we only need to find $\pi_l(\cdot)$ with $g(\pi) \leq 2d$. By using the Frank-Wolfe algorithm under appropriate initialization (Todd, 2016, Proposition 3.17), we can find such a 2-approximate solution with a support smaller than $\xi = 48d \log \log d$.

After that, the server assigns arms in $\mathcal{A}_l$ to agents. Since the server and the agents obtain the same core set when solving the approximate $G$-optimal design[8], the server only needs to send the index among $\xi$ arms to identify and allocate each arm. After pulling arms, agents send the average reward for each arm to the server, who computes the least squares estimator. The agents and the server then eliminate low rewarding arms from their local copies of $\mathcal{A}_l$.

For notational convenience, we define $V(\pi) = \sum_{x \in \mathcal{D}} \pi(x) x x^\top$, $g(\pi) = \max_{x \in \mathcal{D}} x^\top V(\pi)^{-1} x$.

---

**Protocol 2:** Distributed Elimination for Linear Bandits (DELB)

---

1   $\mathcal{A}_1 = \mathcal{D}$, $C_1 = 600$
2   **for** $l = 1, 2, 3, ...$ **do**
     /* All agents and server:  Solve a G-optimal design problem   */
3      Find distribution $\pi_l(\cdot)$ over $\mathcal{A}_l$ such that: 1. its support has size at most $\xi = 48d \log \log d$; 2.
      $g(\pi) \leq 2d$
     /* Server:  Assign pulls and summarize results         */
4      Assign $m_l(x) = \lceil C_1 4^l d^2 \pi_l(x) \ln MT \rceil$ pulls for each arm $x \in \text{Supp}(\pi_l)$ and wait for results[9].
5      Receive rewards for each arm $x \in \mathcal{A}_l$ reported by agents
6      For each arm in the support of $\pi_l(\cdot)$, calculate the average reward $\mu(x)$
7      Compute[10] $X = \sum_{x \in \text{Supp}(\pi_l)} m_l(x) \mu(x) x$, $V_l = \sum_{x \in \text{Supp}(\pi_l)} m_l(x) x x^\top$, $\hat{\theta} = V_l^{-1} X$
8      Send $\hat{\theta}$ to all agents
     /* All agents and server:  Eliminate low-rewarding arms     */
9      Eliminate low rewarding arms: $\mathcal{A}_{l+1} = \left\{ x \in \mathcal{A}_l : \max_{b \in \mathcal{A}_l} \langle \hat{\theta}, b - x \rangle \leq 2^{-l+1} \right\}$

---

## 4.3 REGRET AND COMMUNICATION EFFICIENCY OF DELB

We state our results for the elimination-based protocol for distributed linear bandits. The full proof is given in Appendix G.

**Theorem 3.** *The DELB protocol achieves expected regret $O\left(d\sqrt{TM \log T}\right)$ with communication cost $O\left((Md + d \log \log d) \log T\right)$.*

*Proof sketch:* In round $l$, the number of pulls is at most $48d \log \log d + C_1 4^l d^2 \log MT$. Based on the analysis for elimination-based algorithm, we can show that the suboptimality gap $\langle \theta^*, x^* - x \rangle$ is at most $2^{-l+2}$ with probability $1 - 1/MT$ for any arm $x$ pulled in phase $l$. Suppose there are at most $L$ phases, we can prove that $\mathbb{E}(REG(T)) \leq \sum_{l=1}^{L} O(d\sqrt{4^l d^2 \log^2 TM}) \leq O(d\sqrt{TM \log TM})$.

In each phase, communication cost comes from three parts: assigning arms to agents, receiving average rewards of each arm and sending $\hat{\theta}$ to agents. In the first and second part, each arm $x \in \text{Supp}(\pi_l)$ is designated to as few agents as possible. We can show that the communication cost

---

[8]The Frank-Wolfe algorithm and its initialization are deterministic.

[9]We assign the pulls of each arm to as few agents as possible. See Appendix F for detailed description.

[10]$V_l$ is always invertible when $\mathcal{A}_l$ spans $\mathbb{R}^d$. When $\mathcal{A}_l$ doesn't span $\mathbb{R}^d$, we can always consider $\text{span}(\mathcal{A}_l)$ in phase $l$ and reduce the number of dimensions.

of these parts is $O(M + d \log \log d)$. In the third part, the cost of sending $\hat{\theta}$ is $Md$. Since $l$ is at most $O(\log T)$, the total communication is $O\left((Md + d \log \log d) \log T\right)$. $\qquad\square$

## 4.4 PROTOCOL FOR LINEAR BANDITS WITH TIME-VARYING ACTION SET

In some previous work on linear bandits (Chu et al., 2011; Abbasi-Yadkori et al., 2011), the action set available at timestep $t$ may depend on $t$. That is, players can only choose actions from $\mathcal{D}_t$ at time $t$, while regret is defined against the optimal action in $\mathcal{D}_t$. The DELB protocol does not apply in this scenario. To handle this setting, we propose a different protocol DisLinUCB (Distributed LinUCB) based on LinUCB (Abbasi-Yadkori et al., 2011).

In our distributed algorithm, agent $i$ uses all samples available for him to maintain a confidence set $\mathcal{C}_{t,i} \subseteq \mathbb{R}^d$ for the parameter $\theta^*$. In each step, he chooses an optimistic estimate $\widetilde{\theta}_{t,i} = \text{argmax}_{\theta \in \mathcal{C}_{t-1,i}} \left(\max_{x \in \mathcal{D}} \langle x, \theta \rangle\right)$ and then chooses action $X_{t,i} = \text{argmax}_{x \in \mathcal{D}} \left\langle x, \widetilde{\theta}_{t,i} \right\rangle$, which maximizes the reward according to the estimate $\tilde{\theta}_{t,i}$. We denote $\sum_\tau x_\tau x_\tau^T$ and $\sum_\tau x_\tau y_\tau$ as $W$ and $U$ in our algorithm respectively. We use $W_{t,i}$ and $U_{t,i}$ to denote the sum calculated using available samples for agent $i$ at time step $t$. We construct the confidence set $\mathcal{C}_{t,i}$ using $W_{t,i}$ and $U_{t,i}$:

$$\mathcal{C}_{t,i} = \left\{ \theta \in \mathbb{R}^d : \|\hat{\theta}_{t,i} - \theta\|_{\overline{V}_{t,i}} \leq \sqrt{2 \log \left( \frac{\det\left(\overline{V}_{t,i}\right)^{1/2} \det(\lambda I)^{-1/2}}{\delta} \right)} + \lambda^{1/2} \right\}, \quad (1)$$

where $\overline{V}_{t,i} = \lambda I + W_{t,i}$, $\hat{\theta}_{t,i} = (\lambda I + W_{t,i})^{-1} U_{t,i}$, $\|x\|_V := x^T V x$. Essentially, without synchronization, agents in our protocol execute LinUCB separately. In a synchronization round, agents share all newly acquired samples with each other. The remaining question is when to synchronize.

Our key observation, which comes from the analysis in Abbasi-Yadkori et al. (2011), is that the change in the log-determinant of $\overline{V}_t$ is a good indicator of learning progress. Based on this observation, we only synchronize when agent $i$ finds that the log-determinant of $\overline{V}_{t,i}$ has changed significantly since the last synchronization. We designed this synchronization criterion carefully such that, on the one hand, regret is close to the lower bound and on the other hand, the number of synchronization rounds is small. The full protocol is described below.

---

**Protocol 3:** Distributed Linear UCB (DisLinUCB)

1   **for** $t = 1, \cdots, T$ **do**
2     **for** *Agent* $i = 1, \cdots, M$ **do**
3       Compute $W_{t,i} = \sum_\tau x_\tau x_\tau^T$ and $V_{t,i} = I + \sum_\tau x_\tau y_\tau$ using all available samples
4       Construct the confidence ellipsoid $\mathcal{C}_{t,i}$ using $W_{t,i}$ and $\hat{\theta}_{t,i} = V_{t,i}^{-1} U_{t,i}$.
5       $(x_{t,i}, \tilde{\theta}_{t,i}) = \arg\max_{(x,\theta) \in \mathcal{D}_t \times \mathcal{C}_{t,i}} \langle x, \theta \rangle$
6       Play $x_{t,i}$ and observe reward $y_{t,i}$.
7       Update $W_{t,i}$ and $V_{t,i}$
8       **if** $\log\left(\det V_{t,i}/\det V_{last}\right) \cdot (t - t_{last}) > T \log(MT)/(dM)$ **then**
9         Start a synchronization round
10        $t_{last} = t$, $V_{last} = I + \sum_\tau x_\tau y_\tau$ using all available examples

---

We state the following regret and communication bound for this protocol. The proof is deferred to Appendix H.

**Theorem 4.** *The DisLinUCB protocol can achieve a regret of* $O\left(d\sqrt{MT} \log^2(T)\right)$ *with* $O\left(M^{1.5} d^3\right)$ *communication cost.*

Although the regret bound is still near-optimal, the communication cost has worse dependencies on $M$ and $d$ compared to that of DELB.

## 5 Conclusions and future work

In this work, we propose communication-efficient protocols with near-optimal regret for both multi-armed bandits and linear bandits problem. For multi-armed bandits, we propose DEMAB protocol, which achieves $\sqrt{MKT \log(MK)}$ regret with $M \log(MK)$ communication cost. The communication cost of DEMAB protocol matches the lower bound we proposed except for a logarithmic factor. For distributed linear bandits, our DELB protocol and DisLinUCB protocol achieve near-optimal regret with $O\left((Md + d\log\log d)\log T\right)$ and $O\left(M^{1.5}d^3\right)$ communication cost respectively.

An interesting open question is proving lower bounds of communication cost in order to achieve near-optimal regret in linear bandits. From a reduction from multi-armed bandits to linear bandits, Thm. 2 implies $\Omega(M)$ communication lower bound for linear bandits. However, there is still a gap between this bound and the communication cost of DELB protocol. We conjecture that close-to-$Md$ amount of communication is necessary, which is the case in offline distributed linear regression (Zhang et al., 2013; Braverman et al., 2016) in the context of achieving an optimal risk rate.

## 6 acknowledgements

The authors thank Chi Jin, Chicheng Zhang and Nan Jiang for helpful discussions. This work is supported by National Basic Research Program of China (973 Program) (grant no. 2015CB352502), NSFC (61573026), BJNSF (L172037) and Beijing Acedemy of Artificial Intelligence.

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

# A    ORGANIZATION

The appendix is organized as follows.

Sections B-E are mainly about distributed multi-armed bandits.

In Section B, we present omitted details for the DEMAB protocol. In Section C, we analyze the performance of DEMAB and prove Theorem 1. We also analyze the number of communicated bits in DEMAB in this section. In Section D, we discuss how to remove the usage of public randomness in the DEMAB protocol with little increase in communication cost. In Section E, we give a proof of the communication lower bound for multi-armed bandit.

Sections F-H are mainly about distributed linear bandits.

In Section F, we present the omitted details of DELB protocol. In Section G, we prove the regret and communication cost bounds of DELB protocol, and also analyze the number of communicated bits in DELB. In Section H, we analyze the performance of the DisLinUCB protocol.

Finally, in Section I, we demonstrate that our protocols can be adapted to the P2P communication networks considered in Korda et al. (2016), which proves our claim in footnote 2 of the main paper.

# B  DETAILED DESCRIPTION OF DEMAB

In this section, we give a detailed description of the DEMAB protocol and some subroutines used in the protocol.

---

**Protocol 4:** Distributed Elimination for Multi-armed Bandits (DEMAB)

---

1  $D = \lceil T/MK \rceil$, $C_2 = 1000$, $l_0 = \lfloor \log_4 \left( \frac{D}{C_2 K \log(MK)} \right) \rfloor$, $m_l = \lceil 4^{l+3} \log(MKT \cdot 4^{-l}) \rceil$

   /* Stage 1:  Separate Burn-in                                    */

2  **for** *agent* $i = 1, ..., M$ **do**

3      $\mathcal{A}^{(i)} =$Eliminate$([K],D)$

   /* Switching:  Random Allocation                            */

4  Generate public random numbers $r_1, ..., r_K$ in $[M]$

5  $\mathcal{B}_{l_0+1}^{(i)} = \{a \in \mathcal{A}^{(i)} | r_a = i\}$

   /* Stage 2:  Distributed Elimination                           */

6  **for** $l = l_0 + 1, ...$ **do**

7      **if** *Centralize has not been called* **then**

         /* distributed mode                                          */

8          Agents send $n_l^{(i)} = \left| \mathcal{B}_l^{(i)} \right|$ to server, $N_l = \sum_i n_l^{(i)}$, $N_{max} = \max_i n_l^{(i)}$

9          **if** $N_l \leq M$ **then**

10            Centralize, go to line 22

11         **if** $\vec{n}_l$ *is not balanced* **then**

12            Reallocate

13         Server sends $N_{max}$ to all agents

14         **for** *agent* $i = 1, ..., M$ **do**

15            Pull each $a \in \mathcal{B}_l^{(i)}$ for $m_l$ times, denote average as $\hat{u}_l(\cdot)$

16            Pull other arms in round-robin for $(N_{max} - |\mathcal{B}_l|)m_l$ times

17            Send $(\arg\max_{a'} \hat{u}_l(a'), \max_{a'} \hat{u}_l(a'))$ to server

18         Server receives $(a_{j,l}^*, u_{j,l}^*)$ from agent $j$, and sends $u_l^* = \max_j u_{j,l}^*$ to every agent

19         **for** *agent* $i = 1, ..., M$ **do**

20            Elimination: $\mathcal{B}_{l+1}^{(i)} = \left\{ i \in \mathcal{B}_l^{(i)} : \hat{u}_l(a) + 2^{-l} \geq u_l^* \right\}$

21     **else**

         /* centralized mode                                          */

22         Server assigns arms in $\mathcal{B}_l$ to agents evenly and schedules $m_l$ pulls for each arm

23         Agents pull arms as required by the server, and report the average for the pulled arm

24         Server calculates $\hat{u}_l(a)$, average reward for $m_l$ pulls in this phase, for each arm $a \in \mathcal{B}_l$

25         Elimination: $\mathcal{B}_{l+1} = \left\{ a \in \mathcal{B}_l : \hat{u}_l(a) + 2^{-l} \geq \max_{j \in \mathcal{B}_l} \hat{u}_l(j) \right\}$

---

**Eliminate:** Eliminate executes the single-agent elimination algorithm. In this function, each agent runs the single-agent elimination algorithm for $D$ time steps, then return the remaining arms.

---

**Protocol 5:** Eliminate

---

**Input:** A set of arms $\mathcal{A}_1$, time step $D$.

1  **for** $l = 1, ...$ **do**

2      **for** $a \in \mathcal{A}_l$ **do**

3          Pull arm $a$ for $m_l = \lceil 4^{l+3} \log(4^{-l} MKT) \rceil$ times, denote average reward as $\hat{\mu}_l(a)$

4          If time step $D$ is reached, go to line 6

5          Elimination: $\mathcal{A}_{l+1} = \left\{ a \in \mathcal{A}_l : \hat{\mu}_l(a) > \max_{k \in \mathcal{A}_l} \hat{\mu}_l(k) - 2^{-l} \right\}$

6  Return $\mathcal{A}_l$

---

**Reallocate:** In Reallocate, the server announces the average number of arms; agents with more-than-average arms then donate surplus arms to the server; the server then distributes the donated arms to the other agents, so that every agent has nearly the same number of arms. After calling

Reallocate, $\vec{n}_l$ becomes balanced again. The function contains the following two parts: One running on the server, and the other running on each agent.

---

**Protocol 6:** `Reallocate` for Server

**Input:** $n_l^{(1)},...,n_l^{(M)}$

1   $\bar{n} = \lfloor \sum_{i=1}^{M} n_l^{(i)}/M \rfloor$

2   Send "reallocation", $\bar{n}$ to every player

3   Receive a set of $n_l^{(i)} - \bar{n}$ arms, $\mathcal{A}_i'$, from player $i$ if $n_l^{(i)} > \bar{n}$; $\mathcal{A}_{temp} = \bigcup_i \mathcal{A}_i'$

4   **for** $i = 1,...,M$ **do**

5       If $n_l^{(i)} < \bar{n}$, send $n_l^{(i)} - \bar{n}$ arms in $\mathcal{A}_{temp}$ to player $i$, and remove them from $\mathcal{A}_{temp}$

6   Send the remaining arms in $\mathcal{A}_{temp}$ to players $1,...,|\mathcal{A}_{temp}|$ (one each)

---

**Protocol 7:** `Reallocate` for Agents

1   **if** *Receive "reallocation", $\bar{n}$* **then**

2     **if** $n_l^{(i)} > \bar{n}$ **then**

3         Pick a subset of $n_l^{(i)} - \bar{n}$ arms, $\mathcal{A}_i'$, from $\mathcal{B}_l^{(i)}$, and send them to server

4         $\mathcal{B}_l^{(i)} = \mathcal{B}_l^{(i)} \setminus \mathcal{A}_i'$

5     **else**

6         Wait until receiving $\mathcal{A}_i'$ from server, $\mathcal{B}_l^{(i)} = \mathcal{B}_l^{(i)} \cup \mathcal{A}_i'$

---

**Centralize:** When the number of arms drops below $M$, the subroutine `Centralize` is called, in which agents send their local copy of remaining arms, $\mathcal{B}_l^{(i)}$, to the server, and server receives $\mathcal{B}_l = \bigcup_{i \in [M]} \mathcal{B}_l^{(i)}$.

---

**Protocol 8:** `Centralize`

1   **for** *agent $i = 1,...,M$* **do**

2     Send $\mathcal{B}_l^{(i)}$ to server

3   Server receives $\mathcal{B}_l^{(i)}$ from agent $i$, and calculates $\mathcal{B}_l = \bigcup_i \mathcal{B}_l^{(i)}$

---

**Assignment Strategy:** In centralized mode, server assigns arms to agents in the following way. Let $N_l = |\mathcal{B}_l|$. If $M$ is exactly divisible by $N_l$, for each arm in $\mathcal{B}_l$, server asks $M/N_l$ separate agents to play it for $\lceil m_l N_l/M \rceil$ times. If not, we allocate pulls to agents in the following way: Let $p_l = \lceil m_l N_l/M \rceil$ denote the average pulls each agent needs to perform. Our assignment starts from the arm with the smallest index $a_{k_1}$ and agent 1. For arm $a_{k_j}$ and agent $i$, if agent $i$ has been assigned $p_l$ pulls, we turn to agent $i + 1$. If we have finished allocating $m_l$ pulls for arm $a_{k_j}$, we continue designating arm $a_{k_{j+1}}$. The assignment is finished until all pulls are scheduled to agents.

## C   PROOF OF THEOREM 1

In this section, we give a full proof of Theorem 1, which bounds the total regret and communication cost of the DEMAB protocol. In the analysis below, we will use $\tilde{\mathcal{B}}_l$ to represent $\mathcal{B}_l^{(1)} \cup \cdots \cup \mathcal{B}_l^{(M)}$ (in the distributed mode) or $\mathcal{B}_l$ (in the centralized mode). It refers to the set of remaining arms at the start of the $l$-th phase at stage 2, either held separately by the agents or held by the server.

Suppose that the protocol terminates when $l = l'$. We also let $N_M(a,t) = \sum_{j=1}^{t} \sum_{i=1}^{M} \mathbb{I}[a_{i,j} = a]$ be the number of times arm $a$ is pulled before time step $t$. Without loss of generality, we assume that arm 1 is the best arm, and define $\Delta_k := \mu(1) - \mu(k)$.

We first state a few facts and lemmas.

**Fact 1.** *We state some facts regarding the execution of the algorithm.*

      *1. At line 6 of the server's part in* `Reallocate`, *$|\mathcal{A}_{temp}| < M$;*

2. *After* `Reallocate` *is called,* $\left\langle \left|\mathcal{B}_l^{(1)}\right|, \cdots, \left|\mathcal{B}_l^{(M)}\right| \right\rangle$ *is balanced;*

3. *For any player* $i$*, the number of completed phases in stage 1 is at least* $l_0 = \left\lfloor \log_4 \left( \frac{D}{C_2 K \log MK} \right) \right\rfloor$*, and at most* $l_D = \left\lceil \log_4 \left( \frac{D}{C_2' \log MK} \right) \right\rceil$*.*

4. *The number of phases at stage 2 is less than* $L = 4 + 5\log(MK) = O\left(\log(MK)\right)$*.*

*Proof.* 1. Let $S^+ = \{i : n_l^{(i)} \geq \bar{n}\}$ and $S^- = \{i : n_l^{(i)} < \bar{n}\}$. At line 3 of server's `reallocate` code, server receives $\sum_{i \in S^+} \left( n_l^{(i)} - \bar{n} \right)$ arms. At line 5, $\sum_{i \in S^-} (\bar{n} - n_l^{(i)})$ arms are removed. So at line 6,

$$|\mathcal{A}_{temp}| = \sum_{i \in S^+} (n_l^{(i)} - \bar{n}) - \sum_{i \in S^-} (\bar{n} - n_l^{(i)}) = \sum_{i \in [M]} n_l^{(i)} - M\lfloor \sum_{i \in [M]} n_l^{(i)}/M \rfloor < M.$$

2. Let $n_l^{(i)} = \left|\mathcal{B}_l^{(i)}\right|$. If $\max_i n_l^{(i)} \leq 2\min_i n_l^{(i)}$, the reallocation procedure will do nothing, and $\langle n_l^{(1)}, ..., n_l^{(M)} \rangle$ is by definition balanced. If $\max_i n_l^{(i)} > 2\min_i n_l^{(i)}$, then at the end of the reallocation procedure, every player has a new set of arms $\mathcal{B}_l^{(i)}$ such that $\bar{n} \leq \left|\mathcal{B}_l^{(i)}\right| \leq \bar{n} + 1$. This implies that the number of arms is balanced, since when reallocation is called, $\sum_i n_l^{(i)} \geq M$.

3. The length of the $l$-th phase at stage 1 is at most $K\lceil 4^{l+3} \log(4^{-l} MKT)\rceil$. After $l$ phases, the number of timesteps is at most

$$K \sum_{p=1}^{l} 4^{p+3} \log(4^{-p} MKT) \leq \frac{4^{l+4}}{3} \left( \log(4^{-l} MKT) + \frac{1}{3} \right).$$

Setting $C_2 = 1000$ and $l_0 = \lfloor \log_4 \left( \frac{D}{C_2 K \log MK} \right) \rfloor$, one can verify that

$$\frac{4^{l_0+4}}{3} \left( \log(4^{-l_0} MKT) + \frac{1}{3} \right) \leq \frac{256D}{3C_2 \log(MK)} \left( \log(C_2 M^2 K^3 \log(MK) + \frac{1}{3}) \right) \leq D.$$

Since the total number of timesteps in stage 1 is $D$, we know that the number of completed phases in stage 1 is at least $l_0$.

On the other hand, in $l$ phases, the number of timesteps is at least

$$\sum_{p=1}^{l} 4^{p+3} \log(4^{-p} MKT) \geq 4^{l+3} \log(4^{-l} MKT).$$

By setting $C_2' = 512$ and $l_D = \left\lceil \log_4 \left( \frac{D}{C_2' \log MK} \right) \right\rceil$, one can show that $4^{l_D+3} \log(4^{-l_D} MKT) > D$. Therefore, the number of of completed phases in stage 1 is at most $l_D$.

4. Suppose that stage 2 has $L = 4 + 5\log(MK)$ phases, then the total number of phases during stage 1 and stage 2 is at least $L + l_0$. In phase $L + l_0$, at least $4^{l_0+L+3} \log \left( 4^{-(l_0+L)} MKT \right)$ pulls are made. With some elementary calculation, one can show that the number of arms pulled in phase $L + l_0$ is $MT \cdot \frac{MK}{\log(MK)} \log \frac{\log(MK)}{M}$, which is even greater than the total number of timesteps $MT$. That is to say, the number of phases in stage 2 is less than $L$. $\square$

Via a direct application of Hoeffding's inequality and union bound, we have the following lemma.

**Lemma C.1.** *Denote the average rewards computed by agent* $i$ *in phase* $l$ *of stage 1 be* $\hat{\mu}_{i,l}(\cdot)$*. At stage* 1*, we denote the following even as* $\Lambda_l$*: in phase* $l$ *at stage* 1*, for any agent* $i$*, for any arm* $a \in \mathcal{A}_l^{(i)}$*,*

$$|\hat{\mu}_{i,l}(a) - \mu(a)| \leq 2^{-l-1}.$$

*Then* $\Pr[\Lambda_l] \geq 1 - 2 \cdot 4^l/(MKT)$*.*

*Proof.* For any agent $i$, any phase $l \leq l_D$, denote the empirical mean for arm $a$ in phase $l$ by $\hat{u}_{i,l}(a)$. By a direct application of Hoeffding's bound and union bound, we can observe that for any fixed $i, l, a$, by Hoeffding's bound,

$$\Pr\left[|\hat{\mu}_{i,l}(a) - \mu(a)| > 2^{-l-1}\right] \leq 2\exp\left\{-\frac{1}{2}m_l \cdot 4^{-l-1}\right\} \leq \frac{2 \cdot 4^l}{(MK)^2 T}.$$

A union bound for all arms $a \in \mathcal{A}_l^{(i)}$ and all agents $i \in [M]$ proves the lemma. $\square$

**Lemma C.2.** *For phase $l$ in stage 1, $\Lambda_1 \wedge \cdots \wedge \Lambda_l$ implies the following:*

1. $1 \in \mathcal{A}_{l+1}^{(i)}$;

2. $\forall i \in [M], \forall a \in \mathcal{A}_{l+1}^{(i)}, \mu(a) \geq \mu(1) - 2^{-l+1}$;

3. *Recall that $N_M(k,t) = \sum_{j=1}^{t} \sum_{i=1}^{M} \mathbb{I}[a_{i,j} = k]$ is the number of times arm $k$ is pulled for all agents before time step $t$. If $l_i = \lceil \log_2 \Delta_i^{-1} \rceil + 1 \leq l$, then $N_M(i, D) \leq C_5 M \log(MKT\Delta_i^2)/\Delta_i^2$;*

*In particular, let $l_D$ be the maximum number of phases for all agents at stage 1. $\Lambda_1 \wedge \cdots \wedge \Lambda_{l_D}$ also implies the following:*

4. $\exists i \in [M], 1 \in \mathcal{B}_{l_0+1}^{(i)}$;

5. $\forall i \in [M], \forall a \in \mathcal{B}_{l_0+1}^{(i)}, \mu(a) \geq \mu(1) - 2^{-l_0+1}$;

*Proof.* These results are direct implications of lemma C.1.

1. Notice that if $\Lambda_l$ happens,

$$\hat{\mu}_{i,l}(1) \geq \mu(1) - 2^{-l-1} \geq \mu(a) - 2^{-l-1} \geq \hat{\mu}_{i,l}(a) - 2^{-l}.$$

Thus, arm 1 will never be eliminated throughout the first $l$ phases.

2. $\Lambda_l$ implies that for any $a \in \mathcal{A}_{l+1}^{(i)}$,

$$\hat{\mu}_{i,l}(a) \geq \max_{k \in \mathcal{A}_l^{(i)}} \hat{\mu}_{i,l}(k) - 2^{-l} \geq \hat{\mu}_{i,l}(1) - 2^{-l},$$

which means

$$\mu(a) \geq \hat{\mu}_{i,l}(a) - 2^{-l-1} \geq \hat{\mu}_{i,l}(1) - 2^{-l} - 2^{-l-1} \geq \mu(1) - 2^{-l+1}.$$

3. From statement 2 we know that $\Lambda_1 \wedge \cdots \wedge \Lambda_l$ implies that arm $i$ will be eliminated after $l_i$ phases, which is less than $l$. Therefore, before eliminated, the total number of times arm $i$ is pulled at stage 1 by any agent is at most

$$\sum_{l=1}^{l_i} m_l \leq \frac{4^{l_i+4}}{3}\left(\log(4^{-l_i}MKT) + \frac{1}{3}\right) \leq \frac{C_5 \log(MKT\Delta_i^2)}{\Delta_i^2}$$

for some absolute constant $C_5$. Multiplying by $M$ proves the assertion.

4. Let $\mathcal{A}^{(i)}$ denote the remaining arms at the end of stage 1 for agent $i$. From statement 1 we know that $\Lambda_1 \wedge \cdots \wedge \Lambda_{l_D}$ implies $1 \in \mathcal{A}^{(i)}$ for any $i \in [M]$. So at line 5 in protocol 1, arm 1 will be assign to agent $r_a$.

5. Let $\mathcal{A}^{(i)}$ denote the remaining arms at the end of stage 1 for agent $i$. From statement 2 we know that $\Lambda_1 \wedge \cdots \wedge \Lambda_{l_D}$ implies $\mu(a) \geq \mu(1) - 2^{-l^{(i)}+1}$ for any arm in $\bigcup_{i \in [M]} \mathcal{A}^{(i)}$. Here $l^{(i)}$ denotes the number of phases for agent $i$ at stage 1. Since $\mathcal{B}_{l_0+1} = \bigcup_{i \in [M]} \mathcal{B}_{l_0+1}^{(i)}$ is a subset of $\bigcup_{i \in [M]} \mathcal{A}^{(i)}$ and $l_i \geq l_0$, we conclude that $\forall a \in \mathcal{B}_{l_0+1}, \mu(a) \geq \mu(1) - 2^{-l^{(i)}+1} \geq \mu(1) - 2^{-l_0+1}$.

$\square$

**Lemma C.3.** *For $l > l_0$, denote the following event by $\Omega_l$: at stage 2, for any $a \in \mathcal{B}_l$,*

$$|\hat{u}_l(a) - \mu(a)| \le 2^{-l-1}.$$

*Then $\Pr[\Omega_l] \ge 1 - 2 \cdot 4^l / (M^2 KT)$.*

*Proof.* Recall that for any $l > l_0$, $\hat{u}_l(a)$ is the average of $m_l$ independent samples of the reward from arm $a$. Therefore, for fixed $l$ and $a$, by Hoeffding's inequality,

$$\Pr\left[|\hat{u}_l(a) - \mu(a)| > 2^{-l-1}\right] \le 2 \exp\left\{-\frac{1}{2} m_l \cdot 4^{-l-1}\right\} \le \frac{2 \cdot 4^l}{(MK)^2 T}.$$

A union bound for all $a \in \mathcal{B}_l$ proves this lemma. □

**Lemma C.4.** *Recall that $\Delta_i = \mu(1) - \mu(i)$. If $\Delta_i > 0$, let $l_i = \lceil \log_2 \Delta_i^{-1} \rceil + 1$. Let $\Lambda$ denote $\Lambda_1 \wedge \cdots \wedge \Lambda_{l_D}$. Then at stage 2, $\Lambda \wedge \Omega_{l_0} \wedge \cdots \wedge \Omega_{l_i}$ implies the following:*

$$N_M(i, T) - N_M(i, D) \le \frac{C_4 \log(MKT\Delta_i^2)}{\Delta_i^2} + \eta(a),$$

*where $C_4$ is a universal constant, and $\sum_{a \in [K]} \eta(a) \le M \log M$.*

*Proof.* Suppose that $|\tilde{\mathcal{B}}_l| > M$. By Fact 1, at phase $l \ge l_0 + 1$ in stage 2, the sequence $\left\langle \left|\mathcal{B}_l^{(1)}\right|, \cdots, \left|\mathcal{B}_l^{(M)}\right| \right\rangle$ is balanced. Therefore, during phase $l$ at stage 2, the number of times an arm in $\tilde{\mathcal{B}}_l$ is pulled is at most $2m_l$.

If $|\tilde{\mathcal{B}}_l| \le M$, an arm in $\tilde{\mathcal{B}}_l$ is also pulled at most $2m_l$ times, unless $|\tilde{\mathcal{B}}_l| \cdot m_l < M$. In that case, a phase only lasts for 1 timestep. This is possible only when $m_l < M$, which requires $l < \lfloor \log_4 M \rfloor < \log M$. We denote the number of times $a$ is pulled in such phases by $\eta(a)$.

On the other hand, let $l_i = \lceil \log_2 \frac{1}{\Delta_i} \rceil + 1$. If $l_i \le l_0$, $\Lambda_2$ implies that $i \notin \tilde{\mathcal{B}}_{l_0+1}$. In that case, the number of times arm $i$ is pulled after timestep $D$ is 0. Assume $\Omega_{l_0+1} \wedge \cdots \wedge \Omega_{l_i}$ holds and $i \in \tilde{\mathcal{B}}_{l_i}$. Then

$$\hat{u}_{l_i} \ge \mu(1) - 2^{-l_i-1} \ge \mu(i) + \Delta_i - 2^{-l_i-1} \ge \hat{u}_{l_i}(i) + 2^{-l_i}.$$

Therefore, $\Omega_{l_0+1} \wedge \cdots \wedge \Omega_{l_i}$ implies that $i \notin \tilde{\mathcal{B}}_{l_i+1}$. In this case, number of times arm $i$ is pulled is

$$\begin{aligned}
N_M(i, T) - N_M(i, D) &\le \sum_{l=l_0+1}^{l_i+1} 2m_l + \eta(a) \\
&\le \frac{8}{3} 4^{l_i+4} \left[\log(MKT \cdot 4^{-l_i-1}) + \frac{1}{3}\right] + L + \eta(a) \\
&\le \frac{2^{15} \log(MKT\Delta_i^2)}{3\Delta_i^2} + 4 + 5\log(MK) + \eta(a) \\
&\le \frac{(2^{15} + 27) \log(MKT\Delta_i^2)}{3\Delta_i^2} + \eta(a).
\end{aligned}$$

It is not hard to see that $\sum_{a \in [K]} \eta(a) \le M \log M$. □

**Lemma C.5.** *Let $l \ge l_0 + 1$, and $n_l = \left|\tilde{\mathcal{B}}_l\right|$. With probability $1 - 2LK\delta$, either $n_l < 21M \log \frac{1}{\delta}$ or no reallocation is performed before the start of the $l$-th phase at stage 2.*

*Proof.* Let $Y_{i,l} = \mathbb{I}\left[i \in \tilde{\mathcal{B}}_l\right]$, $X_{i,j} = \mathbb{I}[r_i = j]$, $i \in [K], j \in [M]$. Observe that $X_{i,j}$ and $Y_{i',l}$ are independent. This is because the elimination process between $\tilde{\mathcal{B}}_l$ and $\tilde{\mathcal{B}}_{l+1}$ uses exactly $m_l$

independent samples for each arm; therefore, the conditional probability for remaining has no dependence on which player an arm is assigned to. Let $\vec{Y}$ denote $\{Y_{i,l}, \forall i, l\}$. Since

$$\left|\mathcal{B}_l^{(j)}\right| = \sum_{i=1}^{K} Y_{i,l} X_{i,j}, \ \left|\tilde{\mathcal{B}}_l\right| = n_l = \sum_{i=1}^{K} Y_{i,l},$$

by Chernoff's inequality,

$$\Pr\left[\sum_{i=1}^{K} Y_{i,l} X_{i,j} > \frac{4}{3M}\sum_{i=1}^{K} Y_{i,l}\middle| \vec{Y}\right] \leq \exp\left\{-\frac{\sum_{i=1}^{K} Y_{i,l}}{21M}\right\},$$

$$\Pr\left[\sum_{i=1}^{K} Y_{i,l} X_{i,j} < \frac{2}{3M}\sum_{i=1}^{K} Y_{i,l}\middle| \vec{Y}\right] \leq \exp\left\{-\frac{\sum_{i=1}^{K} Y_{i,l}}{18M}\right\}.$$

Consequently,

$$\Pr\left[\frac{4\left|\tilde{\mathcal{B}}_l\right|}{3M} > \left|\mathcal{B}_l^{(j)}\right| > \frac{2\left|\tilde{\mathcal{B}}_l\right|}{3M}\middle| n_l > 21M\log\frac{1}{\delta}\right]$$

$$\geq 1 - \mathbb{E}_{\vec{Y}}\left[\exp\left(-\frac{n_l}{21M}\right) + \exp\left(-\frac{n_l}{18M}\right)\middle| n_l > 21M\log\frac{1}{\delta}\right]$$

$$\geq 1 - 2\delta.$$

Note that $\frac{2}{3M}\left|\tilde{\mathcal{B}}_l\right| < \left|\mathcal{B}_l^{(j)}\right| < \frac{4}{3M}\left|\tilde{\mathcal{B}}_l\right|$ for all $j \in [K]$ implies that $\left\langle\left|\mathcal{B}_l^{(1)}\right|, \cdots, \left|\mathcal{B}_l^{(M)}\right|\right\rangle$ is balanced. Therefore, if $n_l > 21M\log\frac{1}{\delta}$, with probability $1 - 2(l - l_0)K\delta$, no reallocation will be performed before the $l$-th phase at stage 2. $\square$

Now we are ready to prove Theorem 1.

**Theorem 1.** *The DEMAB protocol incurs $O\left(\sqrt{MTK\log(MK)}\right)$ regret, $O\left(M\log\frac{MK}{\delta}\right)$ communication cost with probability $1 - \delta$, and $O\left(M\log(MK)\right)$ communication cost in expectation.*

*Proof.* **Regret:** Let $l_i = \lceil\log_2\Delta_i^{-1}\rceil + 1$. By Lemma C.2, when $\Lambda_1 \wedge \cdots \wedge \Lambda_{l_i}$ holds,

$$N_M(i, D) \leq \frac{C_5 M\log(MKT\Delta_i^2)}{\Delta_i^2}.$$

Therefore, the probability that this does not hold is at most

$$\sum_{p=1}^{l_i} \Pr\left[\overline{\Lambda_p}\right] \leq \sum_{p=1}^{l_i} \frac{2 \cdot 4^p}{MKT} \leq \frac{128}{3MKT\Delta_i^2}.$$

Thus for any arm $i$,

$$\mathbb{E}\left[N_M(i, D)\right] \leq \frac{C_5 M\log(MKT\Delta_i^2)}{\Delta_i^2} + \frac{128}{3MKT\Delta_i^2} \cdot MD$$

$$\leq \frac{C_5 M\log(MKT\Delta_i^2)}{\Delta_i^2} + \frac{64}{3\Delta_i^2}.$$

Similarly, if $l_i \geq l_0$, when $\Lambda \wedge \Omega_{l_0+1} \wedge \cdots \wedge \Omega_{l_i}$ holds,

$$N_M(i, T) - N_M(i, D) \leq \frac{C_4\log(MKT\Delta_i^2)}{\Delta_i^2} + \eta(i).$$

The probability that this does not hold is at most

$$\Pr\left[\overline{\Lambda}\right] + \sum_{p=l_0}^{l_i+1} \Pr\left[\overline{\Omega_p}\right] \leq \sum_{p=1}^{l_D} \frac{2 \cdot 4^p}{MKT} + \sum_{p=l_0}^{l_i} \frac{2 \cdot 4^p}{M^2 KT} \leq \frac{256\frac{C_2}{C_2'} + \frac{128}{3}}{MT\Delta_i^2} \leq \frac{600}{MT\Delta_i^2}.$$

Therefore

$$\mathbb{E}\left[N_M(i,T) - N_M(i,D)\right] \le \frac{C_4 \log(MKT\Delta_i^2)}{\Delta_i^2} + \eta(i) + \frac{600}{MT\Delta_i^2} \cdot MT$$

$$= \frac{C_4 \log(MKT\Delta_i^2)}{\Delta_i^2} + \frac{600}{\Delta_i^2} + \eta(i).$$

It follows that

$$\sum_{a=1}^{K} \Delta_a \mathbb{E}\left[N_M(a,D)\right] = \sum_{a:\Delta_a > \epsilon_1} \Delta_a \mathbb{E}\left[N_M(a,D)\right] + \sum_{a:\Delta_a \le \epsilon_1} \Delta_a \mathbb{E}[N_M(a,D)]$$

$$\le K \frac{C_5 M \log(MKT\epsilon_1^2) + 256/3}{\epsilon_1} + \epsilon_1 \cdot MD$$

$$\le C_6 \sqrt{MT \log(MK)},$$

where in the last line we choose $\epsilon_1 = \sqrt{\frac{K \log(K)}{T}}$, and $C_6$ is some absolute constant. Similarly,

$$\sum_{a=1}^{K} \Delta_a \mathbb{E}\left[N_M(a,T) - N_M(a,D)\right] \le \sum_{a:\Delta_a > \epsilon_2} \Delta_a \mathbb{E}\left[N_M(a,T) - N_M(a,D)\right] + \epsilon_2 \cdot MT$$

$$\le M \log M + K \frac{C_4 \log(MKT\epsilon_2^2) + C_9}{\epsilon_2} + \epsilon_2 \cdot MT$$

$$\le M \log M + (3C_4 + 601)\sqrt{MKT \log(K)},$$

where in the last line we chose $\epsilon_1 = \sqrt{\frac{K \log(K)}{MT}}$. By our assumption, $M \log M \le \sqrt{MTK \log(K)}$. Therefore by the definition of total regret,

$$\mathbb{E}\left[REG(T)\right] = \sum_{a=1}^{K} \Delta_a \mathbb{E}\left[N_M(a,D)\right] + \sum_{a=1}^{K} \Delta_a \left(\mathbb{E}\left[N_M(a,T)\right] - \mathbb{E}\left[N_M(a,D)\right]\right)$$

$$\le C_6 \sqrt{MT \log(MK)} + (3C_4 + 601 + 1)\sqrt{MKT \log(K)}$$

$$\le \sqrt{2} \cdot \max\{C_6, 3C_4 + 602\}\sqrt{MT(K \log(K) + \log(MK))}.$$

**Communication:** Total communication in stage 1 is 0. We first consider the worst case communication cost in stage 2. Note that during a phase (either in distributed mode or in centralized mode), the communication cost is $O(M)$ if reallocation is not performed. The cost for reallocation at the start of phase $l$ is $O\left(\min\{n_l, n_{l'} - n_l\}\right)$, where $l'$ is the last phase where reallocation is performed. Summing this over all phases (at most $L = O(\log(MK))$) in stage 2, we conclude that total communication cost for reallocation is $O(ML + K)$.

Define $l_1$ to be the first phase such that reallocation is performed. Then, our argument above shows that communication cost is $O(ML + n_{l_1})$. If $n_{l_1} > 21M \log \frac{1}{\delta}$, the event in lemma C.5 will be violated for some $l$. The probability for that is at most $2L^2 K\delta$. By resetting $\delta$, we can show that with probability $1 - \delta$, $n_{l_1} < 21M \log\left(\frac{2L^2 K}{\delta}\right)$. Therefore, with probability $1 - \delta$, total communication cost is

$$O\left(ML + 21M \log\left(\frac{2L^2 K}{\delta}\right)\right) = O\left(M \log \frac{MK}{\delta}\right).$$

In particular, by choosing $\delta = 1/K$, we can show that expected communication cost is $O\left(M \log(MK)\right)$. □

**Theorem 5.** *When $D = l_0 = 0$ in DEMAB, the protocol incurs near-optimal instance-dependent regret $O(\sum_{k:\Delta_k > 0} \Delta_k^{-1} \log T + M \log M)$. With probability $1 - \delta$ the communication cost is $O\left(M \log(T/\delta)\right)$. The expected communication cost is $O\left(M \log T\right)$.*

*Proof.* **Regret:** In the case $D = 0$, it can be seen from the proof of Theorem 1 that

$$\mathbb{E}\left[N_M(i,T)\right] \leq \frac{C_4 \log(MKT\Delta_i^2)}{\Delta_i^2} + \eta(i) + \frac{600}{\Delta_i^2},$$

where $\sum_{i \in [K]} \eta(i) \leq M \log(M)$. It follows that total expected regret is

$$\mathbb{E}\left[\sum_{i:\Delta_i>0} N_M(i,T)\Delta_i\right] \leq \sum_{i:\Delta_i>0} \Delta_i\eta(i) + \sum_{i:\Delta_i>0} \frac{C_4 \log(MKT\Delta_i^2) + 600}{\Delta_i}$$

$$= O\left(M \log(M) + \sum_{k:\Delta_k>0} \Delta_k^{-1} \log T\right).$$

**Communication:** By the proof of Theorem 1, the worst-case communication cost of this protocol is $O\left(M \log T + K\right)$. This is because we need $O(M)$ communication at the end of each phase to perform elimination, and at most a total number of $O(K)$ additional communication among all phases to perform reallocation.

Let $l^*$ be the last complete phase such that $n_{l^*+1} > 21M \log(KL'/\delta)$. By lemma C.5, no reallocation is needed before phase $l^* + 1$ with probability $1 - \delta$, so the total communication before phase $l^* + 1$ is $O(ML')$.

From the beginning of phase $l^* + 1$, the total communication in the following phases is at most $O(ML' + M \log(KL'/\delta))$.

Therefore, with probability $1 - \delta$, the communication cost is $O\left(M \log T + M \log(KL'/\delta)\right) = O\left(M \log(T/\delta)\right)$. Let $\delta = 1/K$, the expected communication is $O\left(M \log T + (ML' + K)/K\right) = O\left(M \log T\right)$.

$\square$

**Finite precision:** We now show that only $O(\log T)$ bits are needed for each number sent in DEMAB. The integers sent in DEMAB are numbers in $\{0, \cdots, K\}$. Therefore $O(\log T)$ bits are sufficient for each integer[11]. In the DEMAB protocol, the only real numbers that are transmitted are the average reward in a phase. In our proof, it is only required that the average of $m_l$ samples is $\frac{1}{4^{l+2} \log(MKT4^{-l})}$-subgaussian. In fact, the average of $m_l$ samples is $\frac{1}{4^{l+3} \log(MKT4^{-l})}$-subgaussian. Therefore, we can use randomized rounding for the average reward with precision $\epsilon = \frac{1}{4TM}$. Then, the rounding error for each real number has zero mean, and is $\frac{1}{M4^{l+3} \log(MKT4^{-l})}$-subgaussian. When computing the average of $m_l$ samples at phase $l$, at most $M$ rounding error terms will contribute to it, whose sum is $\frac{1}{4^{l+3} \log(MKT4^{-l})}$-subgaussian. Therefore, the concentration inequality in lemma C.3 and consequently our main theorem still holds. Apparently $\log_2 \frac{1}{\epsilon} = O\left(\log(MT)\right) = O(\log T)$. Therefore, expected number of communicated bits is $O\left(M \log(MK) \log T\right)$.

## D REMOVING PUBLIC RANDOMNESS

The DEMAB protocol makes use of a public random number generator. It can be viewed as a sequence of uniformly random bits written on a public blackboard that every agent can read, and reading from this sequence does not require communication. In practice, this can be approximated by using a pseudorandom number generator with a truly random seed. In this case, regardless of the amount of public random number used, the communication cost is $O(M)$, which is the cost of broadcasting a short random seed.

However, we can also totally remove the usage of public random numbers. The role of shared randomness in communication complexity has already been investigated. It is known that shared randomness can be efficiently replaced by private randomness and additional communication, as stated by the Newman's Theorem Newman (1991). In our case, the argument is slightly different: we are considering an online learning task instead of function evaluation. Also, in DEMAB, the communication cost itself depends on the public random bits. In particular, we show the following theorem.

---

[11]Recall that $T > K$.

**Theorem 6.** *There exists a protocol for distributed MAB that does not use public randomness with expected regret $O(\sqrt{MKT \log T})$. It has communication cost bounded by $O(M \log(MK) + K)$ (worst case), and expected communication cost $O(M \log(MK))$.*

*Proof.* We make the following modifications to the original DEMAB protocol. Instead of using a public random bit string $s$ [12], we predetermine $B$ strings $s_1,...,s_B$ (which can be hardcoded in advance), and randomly choose from them. That is, the server will generate a random number uniformly distributed in $[B]$, and broadcast it to everyone. The communication cost of doing so will be $M\lceil \log_2 B \rceil$. We now analyze how the choice of $s_1, ..., s_B$ affects the performance of the protocol.

In terms of regret bound, observe that for any random string $s$, the expected regret of any bandits instance $X$ is always $O(\sqrt{MKT \log T})$. Therefore, regret bound will not be affected when public randomness is removed.

Now define $f(X, s)$ to be the expected communication cost of the DEMAB protocol using the public random string $s$ and interacting with the multi-armed bandit instance $X$. Our analysis for DEMAB tells us that $\exists c_1, \forall X$,

$$\mathbb{E}_r\left[f(X, s)\right] \le c_1 M \log(MK).$$

Therefore, if we draw i.i.d. uniform bit strings $s_1, ..., s_B$,

$$\mathbb{E}_{s_1,...,s_B}\left[\frac{1}{B}\sum_{i=1}^{B} f(X, s_i)\right] \le c_1 M \log(MK).$$

We say that a set of bit strings $\{s_1, ..., s_B\}$ is bad for a bandit $X$ if

$$\frac{1}{B}\sum_{i=1}^{B} f(X, s_i) > 2c_1 M \log(MK).$$

We know that there exists $c_2 > 0$ such that $0 \le f(X, s_i) \le c_2(M \log(MK) + K)$. Therefore, by Hoeffding's inequality,

$$\Pr_{s_1,...,s_B}\left[\frac{1}{B}\sum_{i=1}^{B} f(X, s_i) > 2c_1 M \log(MK)\right] \le \exp\left\{-\frac{2Bc_1^2 M \log(MK)}{c_2(1 + K/(M \log(MK)))}\right\}.$$

In other words, for fixed $X$, the probability that we will draw a bad set $\{s_1, ..., s_B\}$ is exponentially small. Therefore, for a family of bandits with size $Q$, the probability for drawing a set of $s_1, ..., s_B$ that is bad for some bandit is at most $Q \cdot \exp\left\{-\frac{2Bc_1^2 M \log(MK)}{c_2(1+K/(M \log(MK)))}\right\}$. If we can show that this quantity is smaller than 1, it would follow that there exists $\{s_1, \cdots, s_B\}$ such that it is not bad for any bandit in the family.

Now, we consider the following family $\mathcal{X}$ of bandits. For each arm, the expected reward could be $a/\Delta$, where $a \in \{0, 1, ..., \lfloor \Delta^{-1} \rfloor\}$. The reward distribution is Bernoulli. The size of this family is $Q \le \left(\Delta^{-1} + 1\right)^K$. Now, consider any other bandit $X_1$. Without loss of generality, we can assume that $X_1$ is a Bernoulli bandit, and that the expectation of each arm is in $[1/4, 3/4]$ [13]. Apparently we can find a bandit $X_2 \in \mathcal{X}$ such that their expected rewards are $\Delta$-close in $\|\cdot\|_\infty$. As a result, the KL-divergence of each arm's reward in $X_1$ and $X_2$ is $O(\Delta^2)$. Let $H(X) = \{a_{1,1}, r_{1,1}, ..., a_{T,M}, r_{T,M}\}$ be the random history of the DEMAB interacting with the bandit instance $X$. Since communication cost is determined given $H(X)$, $\forall s$,

$$|f(X_1, s) - f(X_2, s)| \le c_2\left(M \log(MK) + K\right) d_{TV}(H(X_1), H(X_2))$$
$$= O\left(\left(M \log(MK) + K\right) \cdot \sqrt{TM}\Delta\right).$$

With $\Delta = K^{-1}(MT)^{-0.5}$, the right-hand-side is $O\left(M \log(MK)\right)$. Therefore, it suffices to consider the bandit family $\mathcal{X}$.

---

[12] which has $K\lceil \log_2 M \rceil$ bits

[13] For a general bandit instance $X_1$, when reward $r$ is received, we can generate a Bernoulli reward with expectation $r/2 + 1/4$ to replace it. The regret bound will increase by only a constant factor.

Therefore, we only need to guarantee that $\frac{BM \log(MK)}{1+K/(M \log(MK))} > C'K \log MKT$, where $C'$ is a universal constant. This can be met by setting $B = \lceil 2C'K^2 \log(MKT) \rceil$. In this case, we can guarantee that there exists a set of bit strings $\{s_1, ..., s_B\}$, such that for any bandit instance $X$, when choosing $s$ randomly from this set, the expectation of $f(X, s)$ is $O(M \log(MK))$.

The additional communication overhead for generating the random string (in bits) is

$$O\left(M \log B\right) = O\left(M \log K + M \log \log(MKT)\right).$$

Therefore, under our usual assumption that $T > \max\{M, K\}$, the number of total communicated bits is bounded by $O\left(M \log K + M \log \log T\right)$. In our formulation, we may view $\log T$ bits as one packet. Therefore additional communication cost is $O\left(M\right)$. It follows that total expected communication cost is $O(M \log(MK))$. $\qquad\square$

## E  PROOF FOR THEOREM 2

*Proof.* In order to prove the theorem, we show that for any protocol with communication cost less that $M/c$ ($c$ is a constant), the regret is at least $\Omega(M\sqrt{KT})$.

First, we list two lemmas that will be used in our proof.

**Lemma E.1.** *(Lattimore & Szepesvári, 2019, Theorem 9.1) For $K$-armed bandits, there is an algorithm with expected regret*

$$REG(T) \leq 38\sqrt{KT}.$$

**Lemma E.2.** *(Lattimore & Szepesvári, 2019, Theorem 15.2) For $K$-armed bandits, for any algorithm, there exists an instance such that*

$$REG(T) \geq \frac{1}{75}\sqrt{(K-1)T}.$$

The original lower bound is proved for Gaussian bandits, which doesn't fit exactly in our setting. we modified the proof to work for Bernoulli bandits, which results in a different constant.

We now prove the theorem's statement via a reduction from single agent bandit to multi-agent bandit. That is, we map communication protocols to single-agent algorithms in the following way. For simplicity, we consider protocols as $M$ blocks of code. In agent $i$'s block, each line could be a local computation, sending a message, or waiting for a message to receive.

Consider a communication protocol with communication cost $B(M)$. We denote $X_i$ ($i \in [M]$) to be the number of integers or real numbers that agent $i$ sends or receives throughout a run. $X_i$ is a random variable. Since expected communication cost is less than $M/c$,

$$\sum_{i=1}^{M} \mathbb{E}X_i \leq M/c.$$

Denote $S$ as the set of $M/2$ agents with smaller $\mathbb{E}X_i$. The expected communication cost of any $i \in S$ is at most $2/c$. For any $i \in S$, $\mathbb{P}(X_i \geq 1) \leq \mathbb{E}X_i \leq 2/c$. That is, for any of these agents, the probability of either speaking to or hearing from someone is less than $2/c$. Suppose that agent $j$ is such an agent. Then, we can map the communication protocol to a single-agent algorithm by simulating agent $j$.

The simulation is as follows. Interacting with single agent bandit with time $T$, we run the code for agent $i$ in the protocol. When no communication is needed, we may proceed to the next line of agent $i$'s code. When this line of code sends a message or waits for a message, we terminate the code. In the rest of the timesteps, we run a single-agent optimal algorithm (the one used to realize lemma E.1).

Then, if agent $j$'s code has $\delta$ probability of involving in communication, and if agent $j$'s regret $REG_j(T) \leq A$, via this reduction, we can obtain an algorithm for single-agent MAB with expected regret

$$REG(T) \leq A + \delta \cdot 38\sqrt{KT}.$$

By lemma 2, $REG(T)$ cannot have a regret upper bound better than $\sqrt{T(K-1)}/75$. Therefore

$$A + \delta \cdot 38\sqrt{KT} \geq \sqrt{(K-1)T}/75.$$

If $38\delta < 1/75$, we can show that $A = \Omega\left(\sqrt{KT}\right)$. In our case, $c = 3000$ will suffice. Since we can show this for any agent in $S$, we can show that total regret is $\Omega\left(M\sqrt{KT}\right)$. $\qquad\square$

## F    OMITTED DETAILS OF DELB

**Assignment Strategy:** At line 4, we assign pulls to agents in the following way. Let $p_l = \lceil \sum_x m_l(x)/M \rceil$ denote the average pulls each agent needs to perform. Our assignment starts from the arm with the largest $m_l(x)$ and agent 1. For arm $x_k$ and agent $i$, if agent $i$ has been assigned with $p_l$ pulls, we turn to agent $i+1$. If we have finished designating $m_l(x_k)$ pulls for arm $x_k$, we continue designating arm $x_{k+1}$. The assignment is finished until all pulls are scheduled to agents. Observe that at the start of each phase, each agent has the same $\mathcal{A}_l$ as the server. Therefore, at line 3 they obtain the same $\pi_l$, with the support size at most $d \log \log d$. In that case, the server only needs to send a index ($O(1)$ communication cost) over $\xi = 48d \log \log d$ arms, instead of a vector ($\Omega(d)$ communication cost), to identify an arm $x \in \text{Supp}(\pi_l)$.

## G    PROOF OF THEOREM 3

First, we consider some properties of the elimination based protocol for linear bandits.

**Fact 2.** *Let $T_l$ denote the total number of pulls in the $l$-th phase, then*

$$C_1 4^l d^2 \log MT \leq T_l \leq \xi + C_1 4^l d^2 \log MT,$$

*where $\xi = 48d \log \log d$.*

*Proof.* For an arm $x$ in the core set $\text{Supp}(\pi_l)$, the DELB protocol pulls it $\lceil C_1 4^l d^2 \pi_l(x) \log MT \rceil$ times. Thus the total number of pulls in phase $l$ satisfies

$$\sum_{a \in \text{Supp}(\pi_l)} m_l(a) \geq \sum_{a \in \text{Supp}(\pi_l)} C_1 4^l d^2 \pi_l(x) \geq C_1 4^l d^2 \log(MT),$$

and

$$\sum_{a \in \text{Supp}(\pi_l)} m_l(a) \leq \sum_{a \in \text{Supp}(\pi_l)} \left( C_1 4^l d^2 \pi_l(x) + 1 \right) \leq 48d \log \log d + C_1 4^l d^2 \log MT.$$

$\qquad\square$

**Lemma G.1.** *In phase $l$, with probability $1 - 1/TM$, for any $x \in \mathcal{D}$,*

$$\left| \langle \hat{\theta} - \theta^*, x \rangle \right| \leq 2^{-l}.$$

*Proof.* First, we construct an $\epsilon_l$-covering of $\mathcal{D}$ with $\epsilon_l = 2^{-l-2}$. Denote the center of the covering as $X = \{\bar{x}_1, ..., \bar{x}_Q\}$. Here $Q$ satisfies $Q \leq 3^d 2^{d(l+2)}$.

Assume that $\hat{\theta}$ is calculated from linear regression on $x_1, ... , x_{t'}$. For fixed $x \in \mathcal{D}$, it is known that $\langle \hat{\theta} - \theta^*, x \rangle$ is subgaussian with variance proxy

$$\sum_{s=1}^{t'} \langle x, V_l^{-1} x_s \rangle^2 = \|x\|_{V_l^{-1}}^2 \leq 2\|x\|_{V_l^{-1}}^2. \tag{2}$$

Therefore with probability $1 - 2\delta$,

$$\left| \langle \hat{\theta} - \theta^*, x \rangle \right| \leq 2\sqrt{\|x\|_{V_l^{-1}}^2 \log \frac{1}{\delta}}.$$

Suppose $n_l$ pulls are made in phase $l$. In our case,

$$\|x\|^2_{V_l^{-1}} \le \frac{g(\pi)}{n_l} \le \frac{2}{4^l C_1 d \log MT}.$$

Therefore with probability $1 - 2\delta$,

$$\left|\langle \hat{\theta} - \theta^*, x \rangle\right| \le 2^{-l+1} \sqrt{\frac{2}{C_1 d \log MT} \log \frac{1}{\delta}}.$$

Choose $\delta = 1/(2TMQ)$. It can be shown that with $C_1 = 600$,

$$\frac{2 \log(2MTQ)}{C_1 d \log MT} \le \frac{\log 2 + 1 + d \log 3 + 2d \log 2 + d/2}{300d} \le \frac{1}{64}.$$

Therefore with probability $1 - 1/(TM)$, for all $x \in X$

$$\left|\langle \hat{\theta} - \theta^*, x \rangle\right| \le 2^{-l-2}.$$

Now, consider an arbitrary $x \in \mathcal{D}$. There exists $\bar{x} \in X$ such that $\|x - \bar{x}\| \le 2^{-l-2}$. Therefore with probability $1 - 1/TM$, for any $x \in \mathcal{D}$,

$$\begin{aligned}
\left|\langle \hat{\theta} - \theta^*, x \rangle\right| &\le \left|\langle \hat{\theta} - \theta^*, \bar{x} \rangle\right| + \left|\langle \hat{\theta} - \theta^*, x - \bar{x} \rangle\right| \\
&\le 2^{-l-1} + \|\hat{\theta} - \hat{\theta}^*\| \cdot \|x - \bar{x}\| \\
&\le 2^{-l}.
\end{aligned}$$

$\square$

**Lemma G.2.** *Let $x^* = \arg\max_{x \in D}\langle \theta^*, x \rangle$ be the optimal arm. Then with probability $1 - \log(MT)/(TM)$, $x^*$ will not be eliminated until the protocol terminates.*

*Proof.* If $x^*$ is eliminated at the end of round $l$, one of the following must happen: either (1) $\left|\langle \hat{\theta} - \theta^*, x^* \rangle\right| > 2^{-l}$; or (2) there exists $x \ne x^*$, $\left|\langle \hat{\theta} - \theta^*, x \rangle\right| > 2^{-l}$. Therefore the probability for $x^*$ to be eliminated at a particular round is less than $1 - 1/(TM)$. The total number of phases is at most $\log MT$. Hence a union bound proves the proposition. $\square$

**Lemma G.3.** *Suppose $\delta = 2 \log(TM)/TM$, and $\Delta_x$ denotes the suboptimality gap of $x$, i.e. $\Delta_x = \langle \theta^*, x^* - x \rangle$. For suboptimal $x \in \mathcal{D}$, define $l_x = \inf\{l : 8 \cdot 2^{-l} \le \Delta_x\}$. Then with probability $1 - \delta$, for any suboptimal $x$, $x \notin \mathcal{A}_{l_x}$.*

*Proof.* First, let us only consider the case where $x^*$ is not eliminated. That is,

$$\Pr\left[\exists x \in \mathcal{D} : x \in \mathcal{A}_{l_x}\right] \le \Pr\left[x^* \text{is eliminated}\right] + \Pr\left[\exists x : x \in \mathcal{A}_{l_x - 1}, x \in \mathcal{A}_{l_x} | x^* \in \mathcal{A}_{l_a}\right].$$

Note that conditioned on $x^* \in \mathcal{A}_{l_x}$, $\{x \in \mathcal{A}_{l_x - 1} \wedge x \in \mathcal{A}_{l_x}\}$ implies that at phase $l_x - 1$, either $\left|\langle \hat{\theta} - \theta^*, x \rangle\right| > 2^{-l_x + 1}$ or $\left|\langle \hat{\theta} - \theta^*, x^* \rangle\right| > 2^{-l_x + 1}$. Therefore the probability that there exists such $x$ is less than $\log(TM)/TM$. Hence, with probability $1 - 2\log(TM)/TM$, $x$ will be eliminated before phase $l_x$. $\square$

We are now ready to prove our main result for DELB.

**Theorem 3.** *DELB protocol has expected regret $O\left(d\sqrt{TM \log T}\right)$, and has communication cost $O\left((Md + d \log \log d) \log T\right)$.*

*Proof.* **Regret:** We note that at the start of round $l$, the remaining arms have suboptimality gap at most $8 \cdot 2^{-l}$. Suppose that the last finished phase is $L$. Therefore total regret is

$$\begin{aligned}
REG(T) &\le \sum_{l=1}^{L} C_1 4^l d^2 \log MT \cdot 8 \cdot 2^{-l} + \delta \cdot 2MT \\
&\le C_3 2^L d^2 \log TM.
\end{aligned}$$

Apparently $C_1 4^L d^2 \log TM \leq TM$. Therefore

$$REG(T) \leq \sqrt{C_3^2 4^L d^4 \log^2 TM} \leq C_7 d\sqrt{TM \log TM}.$$

Under our usual assumption that $T > M$, this can be simplified to $O(d\sqrt{TM \log T})$. Here $C_3$ and $C_7$ are some universal constants.

**Communication Cost:** Let $p_l = \sum_x m_l(x)/M$ denote the average pulls each agent needs to perform. Observe that for each arm, the number of agents that it is assigned to is at most $1 + \lceil m_l(x)/p_l \rceil$ agents. Therefore, total communication for scheduling is at most

$$\sum_x \left(\lceil m_l(x)/p_l \rceil + 1\right) \leq 2\xi + M = O(M + d\log\log d).$$

Similarly, total communication for reporting averages is the same. The cost for sending $\hat{\theta}$ is $Md$. Hence, communication cost per phase is $O(Md + d\log\log d)$. On the other hand, total number of phases is apparently $O(\log TM)$. Hence total communication is

$$O\left((Md + d\log\log d)\log TM\right)$$

Under the assumption that $T > M$, this can be simplified to $O\left((Md + d\log\log d)\log T\right)$. $\qquad\square$

**Finite precision:** We now discuss the number of bits needed in DELB. The integers sent in DELB are less than $\max\{T, \xi\}$ (recall that $\xi = 48d\log\log d$). Therefore, every integer can be encoded with $O(\log(dT))$ bits. It remains to be proven that transmitting each real number with logarithmic bits is sufficient. In the DELB protocol, two types of real numbers are transmitted: average of rewards, and entries of $\hat{\theta}$. To transmit real numbers with finite number of bits, we make the following modifications to the original protocol: 1. when transmitting average rewards at line 5, use randomized rounding with precision $\epsilon_1 = \frac{1}{M^2 T}$; 2. after computing $\tilde{\theta} = V_l^{-1} X$ at line 7, let $\hat{\theta}$ be the entry-wise rounded vector of $\tilde{\theta}$ with $\epsilon_2 = \frac{1}{MTd}$.

It can be seen that we only need to prove that after the modifications, lemma G.1 still holds. In each phase, originally $\mu(x)$ is $\frac{1}{m_l(x)}$-subgaussian, but is only required to be $\frac{2}{m_l(x)}$-subgaussian for (2) to hold. After the modification, the contribution of rounding error to a $\mu(x)$ comes from at most $M$ independent terms, and is therefore subgaussian with variance proxy $\frac{1}{MT} \leq \frac{1}{m_l(x)}$. Therefore, after the modifications, the computed $\mu(x)$ is $\frac{2}{m_l(x)}$-subgaussian; hence, (2) holds. It follows that with probability $1 - 1/(TM)$, for all $x \in X$, $\left|\langle\tilde{\theta} - \theta^*, x\rangle\right| \leq 2^{-l-2}$. Therefore, for any $a \in \mathcal{D}$,

$$\left|\langle\tilde{\theta} - \theta^*, x\rangle\right| \leq 2^{-l-1}.$$

Combined with the fact that for any $x \in \mathcal{D}$,

$$\left|\langle\tilde{\theta} - \hat{\theta}, x\rangle\right| \leq \|\tilde{\theta} - \hat{\theta}\| \leq \sqrt{d}\epsilon_2 \leq \frac{1}{MT} \leq 2^{-l-1},$$

we can prove that with probability $1 - 1/(TM)$, for all $x \in X$,

$$\left|\langle\hat{\theta} - \theta^*, x\rangle\right| \leq 2^{-l-2}.$$

Therefore, after the modifications, the regret of the protocol is still $O\left(d\sqrt{TM\log T}\right)$. The amount of communicated bits is thus

$$O\left((Md + d\log\log d)\cdot\log T\cdot\log(dT)\right).$$

# H    PROOF OF THEOREM 4

First, let us restate the DisLinUCB protocol in full detail.

---
**Protocol 3:** Distributed Linear UCB (DisLinUCB)

---
1  $D = T \log(MT)/(dM), \lambda = 1$
2  **for** *Agent $i = 1, ..., M$* **do**
3  $\quad$ Set $W_{syn,i} = 0, U_{syn,i} = 0, W_{new,i} = 0, U_{new,i} = 0, t_{last} = 0, V_{last} = \lambda I$
4  **for** $t = 1, ..., T$ **do**
5  $\quad$ **for** *Agent $i = 1, ..., M$* **do**
6  $\quad\quad$ $\overline{V}_{t,i} = \lambda I + W_{syn,i} + W_{new,i}, \hat{\theta}_{t,i} = \overline{V}_{t,i}^{-1}(U_{syn,i} + U_{new,i})$.
7  $\quad\quad$ Construct the confidence ellipsoid $\mathcal{C}_{t,i}$ using $\overline{V}_{t,i}$ and $\hat{\theta}_{t,i}$.
8  $\quad\quad$ $(x_{t,i}, \tilde{\theta}_{t,i}) = \arg\max_{(x,\theta) \in \mathcal{D}_t \times \mathcal{C}_{t,i}} \langle x, \theta \rangle$
9  $\quad\quad$ Play $x_{t,i}$ and get the reward $y_{t,i}$.
10 $\quad\quad$ Update $W_{new,i} = W_{new,i} + x_{t,i} x_{t,i}^T, U_{new,i} = U_{new,i} + x_{t,i} y_{t,i}$.
11 $\quad\quad$ $V_{t,i} = \lambda I + W_{syn,i} + W_{new,i}$
12 $\quad\quad$ **if** $\log(\det V_{t,i}/\det V_{last,i}) \cdot (t - t_{last}) > D$ **then**
13 $\quad\quad\quad$ Send a synchronization signal to server to start a communication round.
14 $\quad\quad$ **if** *A communication round is started* **then**
15 $\quad\quad\quad$ Send $W_{new,i}$ and $U_{new,i}$ to server
16 $\quad\quad\quad$ Server computes $W_{syn} = W_{syn} + \sum_{j=1}^{M} W_{new,j}, U_{syn} = U_{syn} + \sum_{j=1}^{M} U_{new,j}$
17 $\quad\quad\quad$ Receive $W_{syn}, U_{syn}$ from server
18 $\quad\quad\quad$ Set $W_{new,i} = 0, U_{new,i} = 0, t_{last} = t, V_{last} = \lambda I + W_{syn}$

---

Let us then state several lemmas that will be useful in our proof.

**Lemma H.1.** *For any $\delta > 0$, with probability $1 - M\delta$, $\theta^*$ always lies in the constructed $\mathcal{C}_{t,i}$ for all $t$ and all $i$.*

*Proof.* Using Theorem 2 in Abbasi-Yadkori et al. (2011) and union bound over all agents, we can prove the lemma. $\qquad\square$

For any positive definite matrix $V_0 \in \mathbb{R}^{d \times d}$, any vector $x \in \mathbb{R}^d$, define the norm of $x$ w.r.t. $V_0$ as $\|x\|_{V_0} := \sqrt{x^T V_0 x}$.

**Lemma H.2.** *(Lemma 11 in Abbasi-Yadkori et al. (2011)) Let $\{X_t\}_{t=1}^{\infty}$ be a sequence in $\mathbb{R}^d$, $V$ is a $d \times d$ positive definite matrix and define $\overline{V}_t = V + \sum_{s=1}^{t} X_s X_s^\top$. Then we have that*

$$\log\left(\frac{\det(\overline{V}_n)}{\det(V)}\right) \leq \sum_{t=1}^{n} \|X_t\|_{\overline{V}_{t-1}^{-1}}^2.$$

*Further, if $\|X_t\|_2 \leq L$ for all $t$, then*

$$\sum_{t=1}^{n} \min\left\{1, \|X_t\|_{\overline{V}_{t-1}^{-1}}^2\right\} \leq 2\left(\log\det(\overline{V}_n) - \log\det V\right) \leq 2\left(d\log\left(\left(\text{trace}(V) + nL^2\right)/d\right) - \log\det V\right).$$

Using Lemma H.1, we can bound single step pseudo-regret $r_{t,i}$.

**Lemma H.3.** *With probability $1 - M\delta$, single step pseudo-regret $r_{t,i} = \langle \theta^*, x^* - x_{t,i} \rangle$ is bounded by*

$$r_{t,i} \leq 2\left(\sqrt{2\log\left(\frac{\det(\overline{V}_{t,i})^{1/2}\det(\lambda I)^{-1/2}}{\delta}\right)} + \lambda^{1/2}\right) \|x_{t,i}\|_{\hat{V}_{t,i}^{-1}} = O\left(\sqrt{d\log\frac{T}{\delta}}\right) \|x_{t,i}\|_{\overline{V}_{t,i}^{-1}}.$$
$$(3)$$

*Proof.* Assuming $\theta^* \in \mathcal{C}_{t,i}$,

$$
\begin{aligned}
r_{t,i} &= \langle \theta^*, x^* \rangle - \langle \theta^*, x_{t,i} \rangle \\
&\leq \langle \tilde{\theta}_{t,i}, x_{t,i} \rangle - \langle \theta^*, x_{t,i} \rangle \\
&= \langle \tilde{\theta}_{t,i} - \theta^*, x_{t,i} \rangle \\
&= \langle \tilde{\theta}_{t,i} - \hat{\theta}_{t,i}, x_{t,i} \rangle + \langle \hat{\theta}_{t,i} - \theta^*, x_{t,i} \rangle \\
&\leq \left\| \tilde{\theta}_{t,i} - \hat{\theta}_{t,i} \right\|_{\bar{V}_{t,i}} \|x_{t,i}\|_{\bar{V}_{t,i}^{-1}} + \left\| \hat{\theta}_{t,i} - \theta^* \right\|_{\bar{V}_{t,i}} \|x_{t,i}\|_{\bar{V}_{t,i}^{-1}} \\
&\leq 2 \left( \sqrt{2\log\left( \frac{\det(\bar{V}_{t,i})^{1/2} \det(\lambda I)^{-1/2}}{\delta} \right)} + \lambda^{1/2} \right) \|x_{t,i}\|_{\bar{V}_{t,i}^{-1}} \\
&= O\left( \sqrt{d\log\frac{T}{\delta}} \right) \|x_{t,i}\|_{\bar{V}_{t,i}^{-1}} .
\end{aligned}
$$

$\square$

Now we are ready to prove Theorem 4.

**Theorem 4.** *DisLinUCB protocol achieves a regret of $O\left( d\sqrt{MT}\log^2(T) \right)$ with $O\left( M^{1.5}d^3 \right)$ communication cost.*

*Proof.* **Regret:** Set $\delta = 1/(M^2 T)$, the expected regret caused by the failure of Eq. (3) is at most $MT \cdot 1/(MT) = O(1)$, thus we mainly consider the case where Eq. (3) holds.

In our protocol, there will be a number of epochs divided by communication rounds. We denote $V_{last}$ in epoch $p$ as $V_p$. Suppose that there are $P$ epochs, then $V_P$ will be the matrix with all samples included.

Observe that $\det V_0 = \det(\lambda I) = \lambda^d$. $\det(V_P) \leq \left( \frac{tr(V_P)}{d} \right)^d \leq (\lambda + MT/d)^d$. Therefore

$$
\log\frac{\det(V_P)}{\det(V_0)} \leq d\log\left( 1 + \frac{MT}{\lambda d} \right).
$$

Let $R := \lceil d\log\left(1 + \frac{MT}{\lambda d}\right) \rceil$. It follows that for all but $R$ epochs, we have

$$
1 \leq \frac{\det V_j}{\det V_{j-1}} \leq 2. \tag{4}
$$

We call those satisfying Eq. 4 good epochs. In these epochs, we can use the argument for theorem 4 in Abbasi-Yadkori et al. (2011). First, we imagine the $MT$ pulls are all made by one agent in a round-robin fashion (i.e. he takes $x_{1,1}, x_{1,2},..., x_{1,M}, x_{2,1},..., x_{T,M}$). We use $\tilde{V}_{t,i} = \lambda I + \sum_{\{(p,q):(p<t)\vee(p=t\wedge q<i)\}} x_{p,q}x_{p,q}^T$ to denote the $\overline{V}_{t,i}$ this imaginary agent calculates when he gets to $x_{t,i}$. If $x_{t,i}$ is in one of those good epochs(say the $j$-th epoch), then we can see that

$$
1 \leq \frac{\det \tilde{V}_{t,i}}{\det \bar{V}_{t,i}} \leq \frac{\det V_j}{\det V_{j-1}} \leq 2.
$$

Therefore

$$
\begin{aligned}
r_{t,i} &\leq O\left( \sqrt{d\log\frac{T}{\delta}} \right) \sqrt{x_{t,i}^T \bar{V}_{t,i}^{-1} x_{t,i}} \\
&\leq O\left( \sqrt{d\log\frac{T}{\delta}} \right) \sqrt{x_{t,i}^T \tilde{V}_{t,i}^{-1} x_{t,i} \cdot \frac{\det \tilde{V}_{t,i}}{\det \bar{V}_{t,i}}} \\
&\leq O\left( \sqrt{d\log\frac{T}{\delta}} \right) \sqrt{2x_{t,i}^T \tilde{V}_{t,i}^{-1} x_{t,i}}.
\end{aligned}
$$

We can then use the argument for the single agent regret bound and prove regret in these good epochs.

We denote regret in all good epochs as $REG_{good}$. Suppose $\mathcal{B}_p$ means the set of $(t,i)$ pairs that belong to epoch $p$, and $P_{good}$ means the set of good epochs, using lemma H.2, we have

$$
\begin{aligned}
REG_{good} &= \sum_t \sum_i r_{t,i} \\
&\le \sqrt{MT \sum_{p\in P_{good}} \sum_{(t,i)\in\mathcal{B}_p} r_{t,i}^2} \\
&\le O\left(\sqrt{dMT\log(\frac{T}{\delta}) \sum_{p\in P_{good}} \sum_{(t,i)\in\mathcal{B}_p} \min\left(\|x_{t,i}\|_{\tilde{V}_{t,i}^{-1}}^2, 1\right)}\right) \\
&\le O\left(\sqrt{dMT\log(\frac{T}{\delta}) \sum_{p\in P_{good}} \log\left(\frac{\det(V_p)}{\det(V_{p-1})}\right)}\right) \\
&\le O\left(\sqrt{dMT\log(\frac{T}{\delta}) \log\left(\frac{\det(V_P)}{\det(V_0)}\right)}\right) \\
&\le O\left(d\sqrt{MT}\log(MT)\right).
\end{aligned}
$$

Now we focus on epochs that are not good. For each bad epoch, suppose at the start of the epoch we have $V_{last}$. Suppose that the epoch starts from time step $t_0$, and the length of the epoch is $n$. Then agent $i$ proceeds as $\overline{V}_{t_0,i}, ..., \overline{V}_{t_0+n,i}$. Our argument above tells us that regret in this epoch satisfies

$$
REG \le 2\left(\sqrt{d\log T/\delta}\right) \sum_{i=1}^M \sum_{t=t_0}^n \min\left(\|x_{t,i}\|_{\overline{V}_{t,i}^{-1}}, 1\right) \le O\left(\sqrt{d\log T/\delta}\right) \cdot \sum_{i=1}^M \sqrt{n\log\frac{\det V_{t_0+n,i}}{\det V_{last}}}.
$$

Now, for all but 1 agent, $n\log\frac{\det V_{t_0+n,i}}{\det V_{last}} < D$. Therefore we can show that

$$
REG(n) \le O\left(\sqrt{d\log T/\delta}\right) \cdot M\sqrt{D}.
$$

Since $\det(V_P) \le (\lambda + MT/d)^d$, we know that the number of such epochs are rare. (Less than $R = O(d\log MT)$). Therefore the second part of the regret is

$$
REG_{bad} \le O\left(Md^{1.5}\log^{1.5}MT\right) \cdot D^{1/2}.
$$

If we choose $D = \left(\frac{T\log MT}{dM}\right)$, then $REG(T) = O\left(d\sqrt{MT}\log^2(MT)\right)$. Since $T > M$, we have

$$
REG(T) = O\left(d\sqrt{MT}\log^2(T)\right).
$$

**Communication:** Let $\alpha = \left(\frac{DT}{R}\right)^{0.5}$. Apparently there could be at most $\lceil T/\alpha \rceil$ such epochs that contains more than $\alpha$ time steps. If the $j$-th epoch contains less than $\alpha$ time steps, $\log\left(\frac{\det V_{j+1}}{\det V_j}\right) > \frac{D}{\alpha}$. Since

$$
\sum_{j=0}^{P-1} \log\left(\frac{\det V_{j+1}}{\det V_j}\right) = \log\frac{\det V_P}{\det V_0} \le R,
$$

There could be at most $\lceil\frac{R}{D/\alpha}\rceil = \lceil\frac{R\alpha}{D}\rceil$ epochs with less than $\alpha$ time steps. Therefore, the total number of epochs is at most

$$
\lceil\frac{T}{\alpha}\rceil + \lceil\frac{R\alpha}{D}\rceil = O\left(\sqrt{\frac{TR}{D}}\right).
$$

With our choice of $D$, the right-hand-side is $O\left(M^{0.5}d\right)$. Communication is only required at the end of each epoch, when each agent sends $O(d^2)$ numbers to the server, and then downloads $O(d^2)$ numbers. Therefore, in each epoch, communication cost is $O(Md^2)$. Hence, total communication cost is $O\left(M^{1.5}d^3\right)$. $\qquad\square$

**Finite precision:** we now consider the number of bits transmitted in the DisLinUCB protocol. To that end, we make the following minor modification to DisLinUCB. First, when reward $y_{t,i}$ is observed, we replace it with a random integer in $\{\pm 1\}$ with expectation $y_{t,i}$. Second, after line 8, after $x_{t,i}$ is played, we round each entry of $x_{t,i}$ with precision $\epsilon$, and use the rounded vector in the calculation in line 9. In this case, each entry of $W_{new,i}$ and $U_{new,i}$ is a multiple of $\epsilon^2$. Therefore, transmitting them requires $O(d^2 \log \epsilon^{-1})$ bits. The total communication complexity is then $O\left(M^{1.5} d^3 \log \epsilon^{-1}\right)$ bits.

We now discuss how to choose $\epsilon$ such that regret is not effected. Define

$$\overline{REG}\left(\mathcal{H}\right) := \sum_{i=1}^{M} \sum_{t=1}^{T} \max_{x \in \overline{\mathcal{D}}_t} \langle x - x_{t,i}, \theta^* \rangle,$$

$$REG\left(\mathcal{H}\right) := \sum_{i=1}^{M} \sum_{t=1}^{T} \max_{x \in \mathcal{D}_t} \langle x - x_{t,i}, \theta^* \rangle.$$

Here $\mathcal{H}$ is a shorthand for a history $(x_{1,1}, y_{1,1}, \cdots, x_{T,M}, y_{T,M})$. $\overline{\mathcal{D}}$ refers to set of rounded actions. For every $x \in \mathcal{D}$, there exists $\bar{x} \in \overline{\mathcal{D}}$ such that $\|x - \bar{x}\| \leq \sqrt{d}\epsilon$. Therefore

$$\left| \overline{REG}\left(\mathcal{H}\right) - REG\left(\mathcal{H}\right) \right| \leq MT\sqrt{d}\epsilon.$$

On the other hand, let $\mathcal{H}$ be the (random) history of the DisLinUCB with rounding on action sets $\mathcal{D}_t$, while $\overline{\mathcal{H}}$ is the (random) history of the DisLinUCB with rounding running on action sets $\overline{\mathcal{D}}_t$. Then at each time step, the mapping from past history to the next action is the same. Therefore $KL(\mathcal{H}, \overline{\mathcal{H}}) = O(MTd\epsilon^2)$. It follows that

$$\left| \mathbb{E}\left[\overline{REG}\left(\mathcal{H}\right)\right] - \mathbb{E}\left[\overline{REG}\left(\overline{\mathcal{H}}\right)\right] \right| \leq O\left(\sqrt{M^3 T^3 d}\epsilon\right).$$

When the action set is $\overline{\mathcal{D}}_t$, no rounding is needed, so the regret analysis for the DisLinUCB protocol without rounding directly follows. Therefore,

$$\mathbb{E}\left[\overline{REG}\left(\overline{\mathcal{H}}\right)\right] = O\left(d\sqrt{MT} \log^2 T\right).$$

By choosing $\epsilon = (MT)^{-1}$, we can guarantee $\mathbb{E}\left[REG\left(\mathcal{H}\right)\right] = O\left(d\sqrt{MT} \log^2 T\right)$ for any action set. In this case, the total number of communicated bits is $O\left(M^{1.5} d^3 \log T\right)$.

# I DEMAB AND DELB IN P2P COMMUNICATION NETWORKS

In this section, we will briefly discuss how to implement our protocols (i.e. DEMAB and DELB) in the P2P communication network considered in Korda et al. (2016). We show that our protocols can be adopted to P2P networks after little modification. The communication cost will remain the same, while regret bounds would only increase marginally.

In P2P communication networks, an agent can receive information from at most one other agent at a time, which leads to an information delay for each agent. In order to cope with such delay, we need to extend the length of each communication stage from 1 time step to $M$ time steps, so that agents can complete the communication in turn. Since there are at most $O(\log(MK))$ communication stages in DEMAB and $O(\log T)$ communication stages in DELB, the extension of communication stages incurs at most $O(M^2 \log(MK))$ regret in DEMAB and $O(M^2 \log T)$ regret in DELB for $M$ agents. When the time horizon $T$ is large (i.e. $T > M^3 \log M$), the additional term is dominated by $O\left(\sqrt{MKT \log T}\right)$ and $O\left(d\sqrt{MT \log T}\right)$. Another issue for P2P networks is that there is no longer a physical server in the networks. To solve this problem, we can designate agent 1 as the server: agent 1 will execute both the codes for the server and the codes for an agent.

Specifically, by saying "extending the length of the communication stage", we mean that we can use Procedure 10 and 11 to realize communication subroutines used in our protocols in P2P networks: sending message to the server and receiving messages from the server.

---

**Procedure 4:** Server2Agent: Agent 1 sends message $m_i$ to agent $i$ in a P2P network.

---

**1** For the next $M - 1$ time steps:
    `/* For agent 1:`                                                `*/`
**2**    Send $m_i$ to agent $i + 1$ at the $i$-th step
**3**    Pull an arbitrary arm at each time step

    `/* For agent i(i > 1):`                                 `*/`
**4**    Receive $m_i$ from agent 1 at the $(i - 1)$-st step
**5**    Pull an arbitrary arm at each time step

---

---

**Procedure 5:** Agent2Server: Agents $i(i > 1)$ sends message $m_i$ to agent 1 in a P2P network.

---

**1** For the next $M - 1$ time steps:
    `/* For agent 1:`                                                `*/`
**2**    Receive $m_{i+1}$ from agent $i + 1$ at the $i$-th step
**3**    Pulls an arbitrary arm at each time step

    `/* For agent i(i > 1):`                                 `*/`
**4**    Send $m_i$ to agent 1 at the $(i - 1)$-st step
**5**    Pull an arbitrary arm at each time step

---

### I.1 DEMAB IN P2P NETWORKS

For distributed DEMAB in a P2P network, we can replace the communication stage in DEMAB (i.e. line 8, 12, 15 of Protocol 1) by Procedure Server2Agent and Agent2Server. In this way, it costs $M$ time steps instead of a single time step to collect, aggregate, and boardcast information. We have the following theorem showing the efficacy of distributing DEMAB in a P2P network.

**Theorem 7.** *The DEMAB protocol in P2P networks incurs regret $O\left(\sqrt{MKT \log T} + M^2 \log(MK)\right)$, with expected communication cost $O\left(M \log(MK)\right)$. When $T > M^3 \log M$, the regret bound of this protocol is near-optimal $O\left(\sqrt{MKT \log T}\right)$.*

*Proof.* **Regret:** We compare DEMAB protocol in P2P net with the original one (i.e. Protocol 1). The burn-in stage (i.e. Stage 1) of both protocols are the same. For distributed elimination stage (i.e. Stage 2), the length of each phase in the new protocol is no shorter than that in Protocol 1. Therefore, the number of phases after phase $l_0 + 1$ (included) in new protocol is no more than that in Protocol 1, which is $O(\log(MK))$. In each phase starting from phase $l_0 + 1$, new protocol needs $O(M)$ additional steps to complete the communication in this phase, incurring $O(M)$ additional regret per agent. Therefore, this protocol incurs $O(M^2)$ additional regret per phase starting from phase $l_0 + 1$. The total regret of this protocol is thereby $O\left(\sqrt{MKT \log T} + M^2 \log(MK)\right)$.

**Communication:** We still consider only distributed elimination stage. There are three communication stages per phase in Protocol 1: Line 8, 12, and 15.

In line 8 of DEMAB protocol, the total communication is $O(M)$ since $n_{max}$ is boardcast from the server, and each agent sends $n_l^{(i)}$ to the server. We can observe that the communication cost at corresponding place is also $O(M)$ by replacing the boardcast with Server2Agent. In line 12, the communication cost of both protocols is still the same due to the same reason. In line 15, the new protocol calls Agent2Server which runs for $M$ steps, while the agents report the rewards in the original protocol in a single step. The communication cost is $O(M)$ for both protocols.

In summary, the communication cost of the new protocol is the same as that of Protocol 1, which is $O\left(M \log(MK)\right)$.

$\square$

## I.2 DELB in P2P Networks

Very similar to the P2P version of DEMAB, we can also distribute DELB to P2P networks by replacing the communication stage of DELB by Server2Agent and Agent2Server. We have the following theorem for the P2P version of DELB.

**Theorem 8.** *The DELB protocol in a P2P network has regret* $O\left(d\sqrt{MT\log T} + M^2\log T\right)$ *with expected communication cost* $O\left((Md + d\log\log d)\log T\right)$. *When* $T > M^3\log M$, *the regret of DELB is a near-optimal regret* $O\left(d\sqrt{MT\log T}\right)$.

*Proof.* The proof is very similar to the proof of theorem 7. Note that in the P2P version of DELB, there are $O(\log T)$ communication stages in total, which incurs $O(M^2\log T)$ additional regret. The communication cost of the new protocol is the same as Protocol 2 for the same reason mentioned in the proof of theorem 7. □

