# OpenReview forum: "Distributed Bandit Learning: Near-Optimal Regret with Efficient Communication"
_ICLR.cc/2020/Conference — Accept (Poster)_

### Official Review · AnonReviewer3 · 2019-10-23
**Official Blind Review #3**

**Rating:** 6

**Review:**

The paper considers the problem of distributed multi-arm bandit, where M players are playing in the same stochastic environment. The goal of the paper is to have small over-all regret for all the players without a significant amount of communication between the players.



The main contribution of this paper is obtaining regret ~root(M KT) with ~M bits of communication in MAB, and regret ~d*root(MT) with ~Md bits of communication in linear bandit setting.

The main intuition of the algorithms in this paper is to do "best arm identification" with epoching: At every epoch t, the central server sends the set of possible best arms to each player and each player pulls it for 2^t /M times, followed by a communication round. Thus, the cumulative regret is comparable to having one player doing this epoch strategy for MT iterations, where the regret follows.

The problem considered in this paper is interesting and the result is new, the technique looks simple on paper but it requires a masterful combination of known tricks in (linear) MAB to obtain the best bound.


It seems that in the MAB setting, the lower bound could be further strengthened with a log(K) factor, since removing this factor would ultimately require "dynamic epoching" which is not possible with limited communication. This would mostly complete the picture in the distributed MAB regime.


Missing citation:
The authors are missing citations relevant to distributed MAB with collisions, see for example
"Non-Stochastic Multi-Player Multi-Armed Bandits: Optimal Rate With Collision Information, Sublinear Without"


After Rebuttal: I have read the authors' responses and acknowledge the sensibility of the statement.




**Experience Assessment:**

I have published in this field for several years.

**Review Assessment: Checking Correctness Of Derivations And Theory:**

I assessed the sensibility of the derivations and theory.

**Review Assessment: Checking Correctness Of Experiments:**

N/A

**Review Assessment: Thoroughness In Paper Reading:**

I read the paper at least twice and used my best judgement in assessing the paper.

---

> ### Author Response · Authors · 2019-11-10
> **Response to Reviewer 3**
>
> We thank anonymous reviewer 3 for the review.
>
> Regarding the lower bound: We greatly thank reviewer 3 for the comments on the lower bound. We are considering this and may improve the lower bound in the final version.
>
> Regarding the missing citation: Thanks for bringing this paper to our attention. We will add it to the references.

---

### Official Review · AnonReviewer1 · 2019-10-25
**Official Blind Review #1**

**Rating:** 6

**Review:**

The authors study a bandit problem where there are multiple agents (say, M) and each of the agents is playing a multi-armed bandit problem for T rounds. The agents can communicate with each other in order to achieve small regret. The problem is to design a strategy for arm-playing and communication so that the agents all combined can achieve a small regret w/o communicating a lot. The authors study this bandit problem in 2 settings: 1. multi-armed bandit setting and 2.  bandit linear optimization. For both these settings, the authors establish elimination style algorithms with communication. upon communication the sub-optimal arms are eliminated and the game continues with the remaining arms.
The authors establish regret guarantees as well as communication guarantees. The interesting result is that with constant communication the regret scales as if full communication was available.

The results are interesting and I do not have any objections with the paper, except that ICLR might not be the right avenue for such work given that it lacks any ideas regarding representation learning.

**Experience Assessment:**

I have published one or two papers in this area.

**Review Assessment: Checking Correctness Of Derivations And Theory:**

I did not assess the derivations or theory.

**Review Assessment: Checking Correctness Of Experiments:**

N/A

**Review Assessment: Thoroughness In Paper Reading:**

I made a quick assessment of this paper.

---

> ### Author Response · Authors · 2019-11-10
> **Response to Reviewer 1**
>
> We thank anonymous reviewer 1 for the review.
>
> Regarding the ICLR venue: Our work focuses on parallelization and distributed learning, which is also a research topic in representation learning. We believe our results can bring insight to other distributed learning problems. Besides, the linear bandit studied in our paper is a special case of contextual bandit. We notice that several contextual bandit papers have been published on ICLR in the past a few years. Extending our work to contextual bandit is a future direction.

---

### Official Review · AnonReviewer2 · 2019-10-27
**Official Blind Review #2**

**Rating:** 8

**Review:**


This paper considers the problem of obtaining an optimal regret algorithm in a distributed setting without incurring a large communication cost. In the standard and linear MAB settings, the authors propose algorithms and show that they achieve optimal regret up to logarithmic factors with communication costs that are almost independent of the horizon T. In addition the authors establish interesting lower bounds on the communication cost to obtain sublunar regret.

Overall I found the paper very well motivated and clearly written. I did not go through the proofs in the appendix carefully, but I did check a few sections and found them to be correct. I did have a few concerns.

I found the discussion around load balancing very confusing. Perhaps the authors could provide a picture to explain the issue? In addition, there is a lack of experiments- it is always nice to see comparisons to baseline even though the theory implies you would do better. Finally, the algorithm doesn’t seem extremely practical from an applied point of view - for a linear amount of time all the bandits are pulling the same arms and there is no communication at all. I understand this repeated work doesn’t affect the regret - but it is an artifact of using elimination. In general optimism based approaches (such as UCB) tend to work significantly better than elimination-style methods. I’d be curious to hear the authors comment on whether they think a UCB style algorithm is possible in this setting.

Overall I recommend the paper for acceptance.




**Experience Assessment:**

I have published one or two papers in this area.

**Review Assessment: Checking Correctness Of Derivations And Theory:**

I assessed the sensibility of the derivations and theory.

**Review Assessment: Checking Correctness Of Experiments:**

N/A

**Review Assessment: Thoroughness In Paper Reading:**

I read the paper at least twice and used my best judgement in assessing the paper.

---

> ### Author Response · Authors · 2019-11-10
> **Response to Reviewer 2**
>
> We thank anonymous reviewer 2 for the review.
>
> Regarding the load balancing: The main issue of load balancing is that the length of each phase is determined by the agent with maximum number of remaining arms, since all agents run concurrently and need to synchronize at the end of each epoch. As a result, agents with too many arms need to donate arms to other agents if the numbers of arms remained in agents are not balanced. We will make it clearer in the final version.
>
> Regarding the experiments: We tested the algorithms for distributed linear bandits. We compared performance of our algorithms with two baselines: LinUCB with no communication (LinUCBnoComm) and adapted LinUCB with naive communication strategy (LinUCBnaiveComm), which means we allow agents to communicate their full information at some time steps. In our experiment, we enforced a communication budget of 200kb. The result shows that the regret of DELB protocol is 4 times smaller than that of LinUCBComm or LinUCBnaiveComm. In detail, we ran the algorithms using M = 100 agents on synthesized datasets with 1000 actions, d = 10. We used T ranging from 100 to 10000. We will update the final version accordingly.
>
> Regarding the UCB algorithm: In fact, for multi-armed bandits, the first protocol we designed with near-optimal regret and efficient communication is based on UCB approach. The main idea is to maintain global counts for pulling each arms and synchronize the average rewards for a certain arm only when the number of pulling that arm is doubled. However, the communication cost of this UCB-based protocol is $O(MK\log M)$, which is worse than DEMAB. For linear bandits, our DisLinUCB protocol is based on optimism principle, which also achieves near-optimal regret with efficient communication.

---

### Decision · Program_Chairs · 2019-12-19

**Decision:**

Accept (Poster)

**Comment:**

This paper tackles the problem of regret minimization in a multi-agent bandit problem, where distributed learning bandit algorithms collaborate in order to minimize their total regret. More specifically, the work focuses on efficient communication protocols and the regret corresponds to the communication cost. The goal is therefore to design protocols with little communication cost. The authors first establish lower bounds on the communication cost, and then introduce an algorithm with provable near-optimal regret.

The only concern with the paper is that ICLR may not be the appropriate venue given that this work lacks representation learning contributions. However, all reviewers being otherwise positive about the quality and contributions of this work, I would recommend acceptance.